# Nearly Optimal Best-of-Both-Worlds Algorithms for Online Learning with Feedback Graphs

**Shinji Ito**
NEC Corporation, Tokyo, Japan
RIKEN AIP, Tokyo, Japan
i-shinji@nec.com

**Taira Tsuchiya**
Kyoto University, Kyoto, Japan
RIKEN AIP, Tokyo, Japan
tsuchiya@sys.i.kyoto-u.ac.jp

**Junya Honda**
Kyoto University, Kyoto, Japan
RIKEN AIP, Tokyo, Japan
honda@i.kyoto-u.ac.jp

## Abstract

This study considers online learning with general directed feedback graphs. For this problem, we present best-of-both-worlds algorithms that achieve nearly tight regret bounds for adversarial environments as well as poly-logarithmic regret bounds for stochastic environments. As Alon et al. [2015] have shown, tight regret bounds depend on the structure of the feedback graph: *strongly observable* graphs yield minimax regret of $\tilde{\Theta}(\alpha^{1/2}T^{1/2})$, while *weakly observable* graphs induce minimax regret of $\tilde{\Theta}(\delta^{1/3}T^{2/3})$, where $\alpha$ and $\delta$, respectively, represent the independence number of the graph and the domination number of a certain portion of the graph. Our proposed algorithm for strongly observable graphs has a regret bound of $\tilde{O}(\alpha^{1/2}T^{1/2})$ for adversarial environments, as well as of $O(\frac{\alpha(\ln T)^3}{\Delta_{\min}})$ for stochastic environments, where $\Delta_{\min}$ expresses the minimum suboptimality gap. This result resolves an open question raised by Erez and Koren [2021]. We also provide an algorithm for weakly observable graphs that achieves a regret bound of $\tilde{O}(\delta^{1/3}T^{2/3})$ for adversarial environments and poly-logarithmic regret for stochastic environments. The proposed algorithms are based on the follow-the-regularized-leader approach combined with newly designed update rules for learning rates.

## 1 Introduction

In this paper, we consider *online learning with feedback graphs* [Mannor and Shamir, 2011], a common generalization of the multi-armed bandit problem [Lai et al., 1985, Auer et al., 2002a,b] and the problem of prediction with expert advice [Littlestone and Warmuth, 1994, Freund and Schapire, 1997]. This problem is a sequential decision-making problem formulated with a directed *feedback graph* $G = (V, E)$, where $V = [K] := \{1, 2, \ldots, K\}$ is the set of arms or available actions, and $E \subseteq V \times V$ represents the structure of feedback for choosing actions. In each round of $t = 1, 2, \ldots, T$, a player sequentially chooses an action $I_t \in V$ and then incurs the loss of $\ell_t(I_t)$, where $\ell_t : V \to [0, 1]$ is a loss function chosen by the environment. After choosing the action, the player gets feedback of $\ell_t(j)$ for all $j$ such that the feedback graph $G$ has an edge from $I_t$ to $j$. If $G$ consists of only self-loops, i.e., if $E = \{(i, i) \mid i \in V\}$, the problem corresponds to a $K$-armed bandit problem. If $G$ is a complete directed graph with self-loops, i.e., $E = V \times V$, then the problem corresponds to a problem of prediction with expert advice.

Alon et al. [2015] have provided a characterization of minimax regrets for the problem of online learning with feedback graphs. They divide the class of all directed graphs into three categories. For the first category, called *strongly observable graphs*, the minimax regret is $\tilde{\Theta}(\alpha^{1/2}T^{1/2})$, where $\alpha$ is the independence number of the graph $G$, and $\tilde{\Theta}$ ignores poly-logarithmic factors in $T$ and $K$. For the second category, *weakly observable graphs*, the minimax regret is $\tilde{\Theta}(\delta^{1/3}T^{2/3})$, where $\delta$ represents the *weakly dominating number*. For the last category of *unobservable graphs*, it is not possible to achieve sublinear regret, which means that the minimax regret is $\Theta(T)$. The definitions of categories of graphs and $\alpha$ and $\delta$ are given in Section 3.

Best-of-both-worlds (BOBW) algorithms [Bubeck and Slivkins, 2012] have been studied for the purpose of going beyond such minimax regret bounds; they achieve sublinear regret for adversarial environments and, as well, have logarithmic regret bounds for stochastic environments. The only BOBW algorithm for online learning with feedback graphs has been proposed by Erez and Koren [2021]. They have focused on the case in which $G$ is symmetric and all vertices have self-loops, i.e., any edge $(i,j) \in E$ is accompanied by its reversed edge $(j,i) \in E$ and $(i,i) \in E$ for any $i \in V$. Note that this is a special case of strongly observable graphs. For this class of problems, they provide an algorithm that achieves a regret bound of $\tilde{O}(\theta^{1/2}T^{1/2})$ for adversarial environments, and of $O\left(\frac{\theta \mathrm{polylog}(T)}{\Delta_{\min}}\right)$ for stochastic environments, where $\theta$ ($\geq \alpha$) is the *clique covering number* of the graph $G$, and $\Delta_{\min}$ is the minimum *suboptimality gap* for the loss distributions. Their algorithm also works well for adversarially-corrupted stochastic environments, achieving $O\left(\frac{\theta \mathrm{polylog}(T)}{\Delta_{\min}} + \left(\frac{C\theta \mathrm{polylog}(T)}{\Delta_{\min}}\right)^{1/2}\right)$-regret, where $C$ represents the total amount of corruption.

As Erez and Koren [2021] have pointed out, however, their results leave room for improvement, which is due to the fact that the clique covering number $\theta$ is significantly larger than the independence number $\alpha$ in some cases. Indeed, there is an example such that $\alpha = 1$ while $\theta = K$, as mentioned in Section 3. This means that regret bound depending on $\theta$ is not minimax optimal. In response to this issue, they have raised the question of whether it is possible to replace $\alpha$ with $\theta$ in their regret bounds. Contributions of this study include a positive solution to this question.

## 1.1 Contributions of this study

This study provides BOBW algorithms that achieve minimax regret (up to logarithmic factors) for online learning with general feedback graphs. Our contributions can be summarized as follows:

**Theorem 1** (strongly observable case, informal)**.** *For the problem with strongly observable graphs, an algorithm achieves $R_T = \tilde{O}(\alpha^{1/2}T^{1/2})$ for adversarial environments, $R_T = O\left(\frac{\alpha(\ln T)^3}{\Delta_{\min}}\right)$ for stochastic environments, and $R_T = O\left(\frac{\alpha(\ln T)^3}{\Delta_{\min}} + \left(\frac{C\alpha(\ln T)^3}{\Delta_{\min}}\right)^{1/2}\right)$ for adversarially-corrupted stochastic environments, where $\alpha$ is the independence number of feedback graphs.*

**Theorem 2** (weakly observable case, informal)**.** *For the problem with weakly observable graphs, an algorithm achieves $R_T = \tilde{O}(\delta^{1/3}T^{2/3})$ for adversarial environments, $R_T = O\left(\frac{\delta(\ln T)^2}{\Delta_{\min}^2} + \frac{K'\ln T}{\Delta_{\min}}\right)$ for stochastic environments, and $R_T = O\left(\frac{\delta(\ln T)^2}{\Delta_{\min}^2} + \left(\frac{C^2\delta(\ln T)^2}{\Delta_{\min}^2}\right)^{1/3} + \frac{K'\ln T}{\Delta_{\min}} + \left(\frac{CK'\ln T}{\Delta_{\min}}\right)^{1/2}\right)$ for adversarially-corrupted stochastic environments, where $\delta$ is the weakly dominating number of feedback graphs, and $K'(\leq K)$ is the number of vertices not covered by the weakly dominating set.*

**Remark 1.** The regret bound in Theorem 2 for stochastic environments include an $O\left(\frac{K'\ln T}{\Delta_{\min}}\right)$-term, which is negligibly small compared to the other term $\frac{\delta(\ln T)^2}{\Delta_{\min}^2}$ when $T$ is sufficiently large. However, if $K'$ is larger than $\frac{\delta\ln T}{\Delta_{\min}}$, this term can be dominant. In such a case, the regret upper bound may be improved by modifying the algorithm. Roughly speaking, by combining the approach to strongly observable case, the $O\left(\frac{K'\ln T}{\Delta_{\min}}\right)$-term can be replaced with an $O\left(\frac{\alpha'(\ln T)^3}{\Delta_{\min}}\right)$-term, where $\alpha'$ is the independence number of the subgraph consisting of vertices not dominated by the weakly dominating set. If $\alpha'(\ln T)^2 \leq K'$, the modified version provides a better regret bound. Details of the modification are given Appendix C.

Table 1: Regret upper bounds for online learning with feedback graphs. Note that regret bounds by Erez and Koren [2021] and Rouyer et al. [2022] only apply to a special case of strongly observable graphs with self-loops. We also note that the graph consisting only of self-loops, which corresponds to the standard multi-armed bandit problem, is a special case of strongly observable graphs.

| feedback graph | reference | adversarial | stochastic |
| --- | --- | --- | --- |
| strongly observable | [Alon et al., 2015] | $\tilde{O}\left(\alpha^{1/2}T^{1/2}\right)$ | $\tilde{O}\left(\alpha^{1/2}T^{1/2}\right)$ |
| | [Erez and Koren, 2021] | $\tilde{O}\left(\theta^{1/2}T^{1/2}\right)$ | $O\left(\sum_k \frac{(\ln T)^4}{\Delta_k}\right)$ |
| | [Rouyer et al., 2022] | $\tilde{O}\left(\alpha^{1/2}T^{1/2}\right)$ | $O\left(\sum_{i\in S} \frac{(\ln T)^2}{\Delta_i}\right)$ |
| | **[This work]** Theorem 1 | $\tilde{O}\left(\alpha^{1/2}T^{1/2}\right)$ | $O\left(\frac{\alpha(\ln T)^3}{\Delta_{\min}}\right)$ |
| self-loops only (standard MAB) | [Zimmert and Seldin, 2021] | $O\left(K^{1/2}T^{1/2}\right)$ | $O\left(\sum_{i:\Delta_i>0} \frac{\ln T}{\Delta_i}\right)$ |
| | **[This work]** Theorem 1 | $\tilde{O}\left(K^{1/2}T^{1/2}\right)$ | $O\left(\frac{K(\ln T)^3}{\Delta_{\min}}\right)$ |
| weakly observable | [Alon et al., 2015] | $\tilde{O}\left(\delta^{1/3}T^{2/3}\right)$ | $\tilde{O}\left(\delta^{1/3}T^{2/3}\right)$ |
| | [Kong et al., 2022] | $\tilde{O}\left(K^{2/3}\delta^{1/3}T^{2/3}\right)$ | $O\left(\delta^2\frac{(\ln T)^{3/2}}{\Delta_{\min}^3}\right)$ |
| | **[This work]** Theorem 2 | $\tilde{O}\left(\delta^{1/3}T^{2/3}\right)$ | $O\left(\frac{\delta(\ln T)^2}{\Delta_{\min}^2} + \frac{K'\ln T}{\Delta_{\min}}\right)$ |

Regret bounds for online learning with feedback graphs are summarized in Table 1. Note that the regret bounds by Erez and Koren [2021] apply only to the special case of strongly observable graphs that have self-loops for all vertices. Their algorithm and regret bounds are stated with *clique cover* $\{V_k\}_{k=1}^{L}$ of $G$, which is a partition of all vertices $V$ such that each $V_k$ is a clique. The clique covering number $\theta$ of $G$ is the minimum size $L$ of clique covers. Parameters $\Delta_k$ in Table 1 are defined to be the minimum suboptimality gap among actions in $V_k$, and the summation is taken over $k \in [L]$ such that $\Delta_k > 0$. As the clique covering number $\theta$ is larger than or equal to the independence number $\alpha$ of $G$, our adversarial regret bound of $\tilde{O}\left(\alpha^{1/2}T^{1/2}\right)$ for strongly observable cases is superior to that obtained by Erez and Koren [2021] and is minimax optimal up to logarithmic factors. Although our stochastic regret bound is also better than one by Erez and Koren [2021] in many cases, it is not always so. For example, if $\alpha = \theta$ and $\Delta_{\min}$ is much smaller than many $\Delta_k$, their regret may be better. Note that the work by Rouyer et al. [2022], which proposes BOBW algorithms for strongly observable graphs with self-loops, has been published at NeurIPS 2022, independently of this study. While their algorithms achieve better regret bounds for a certain class of problem settings, our results have the advantage of being applicable to a wider range of problem settings, including directed feedback graphs without self-loops and adversarially corrupted stochastic settings. A more detailed discussion can be found in Appendix B

Our study includes the first nearly optimal BOBW algorithm that can be applied to online learning with weakly observable graphs. As shown in Table 1, the adversarial regret bounds obtained with the proposed algorithm match the minimax regret bound shown by Alon et al. [2015], up to logarithmic factors. Similarly to their algorithm, the proposed algorithm uses a *weakly dominating set* $D \subseteq V$ of $G$. If $D$ is a weakly dominating set, all elements in the set of vertices not dominated by $D$, which is denoted by $V_2 \subseteq V$, have self-loops. Parameters $\delta$ and $K$ in the regret bounds are given by $\delta = |D|$ and $K' = |V_2|$. The stochastic regret bound obtained with the proposed algorithm is also nearly tight. In fact, Alon et al. [2015] have shown a regret lower bound of $\tilde{\Omega}\left(\frac{\delta}{\Delta_{\min}^2}\right)$ in the proof of Theorem 7 in their paper. Further, if vertices in $V_2$ are not connected by edges except for self-loops, the problem is at least harder than the $K'$-armed bandit problem, which leads to a regret lower bound of $\Omega\left(\frac{K'\ln T}{\Delta_{\min}}\right)$. We note that, just before the submission of this paper, Kong et al. [2022] published a work on BOBW algorithms applicable to weakly observable graphs, of which regret bounds are also included in Table 1.

## 1.2 Techniques employed in this study

The proposed algorithms are based on the follow-the-regularized-leader (FTRL) framework, similarly to the algorithms by Alon et al. [2015] and Erez and Koren [2021]. The main differences with existing methods are in the definitions of regularization functions and update rules for learning rates.

For strongly observable cases, we employ the Shannon entropy regularizer functions with a newly developed update rule for learning rates. Most FTRL-based BOBW algorithms are realized by setting the learning rate adaptively to $t$ and/or observations. On the other hand, it is well known that FTRL with Shannon-entropy regularization corresponds to Exp3 algorithm [Auer et al., 2002b] as discussed in, e.g., Lattimore and Szepesvári [2020, Example 28.3]. Since Exp3.G by Alon et al. [2015] achieves an independence-number-dependent regret bound for adversarial environments, it is intuitively natural to expect that a variant of Exp3.G with adaptive learning rates can be used to achieve BOBW regret bounds. However, from the theoretical viewpoint, it is necessary to express the regret depending on the arm-selection distribution $q_t$ to apply the *self-bounding technique* [Gaillard et al., 2014, Zimmert and Seldin, 2021], which plays the central role in the BOBW analysis.

The proposed algorithm for weakly observable graphs uses novel regularization functions consisting of Tsallis-entropy-based and Shannon-entropy-based regularization. Intuitively, we divide the vertices $V$ into the weakly dominated part $V_1$ and non-dominated part $V_2$, and apply Shannon-entropy regularization to $V_1$ and Tsallis-entropy regularization to $V_2$. We combine the FTRL method with exploration using a uniform distribution over the weakly dominating set, similarly to the approach by Alon et al. [2015]. However, we adjust exploration rates and learning rates in a carefully designed manner, in contrast to the existing approach that employs fixed parameters. The combination of the above techniques leads to an entropy-dependent regret bound. By applying the self-bounding technique to this bound, we obtain improved regret bounds for stochastic environments.

## 2 Related work

Since Bubeck and Slivkins [2012] initiated the study of best-of-both-worlds (BOBW) algorithms for the multi-armed bandit (MAB) problem, studies on BOBW algorithms have been extended to a variety of problem settings, including the problem of prediction with expert advice [Gaillard et al., 2014, Luo and Schapire, 2015], combinatorial semi-bandits [Zimmert et al., 2019, Ito, 2021a], linear bandits [Lee et al., 2021], episodic Markov decision processes [Jin and Luo, 2020, Jin et al., 2021], bandits with switching costs [Rouyer et al., 2021, Amir et al., 2022], bandits with delayed feedback [Masoudian et al., 2022], online submodular optimization [Ito, 2022], and online learning with feedback graphs [Erez and Koren, 2021, Kong et al., 2022, Rouyer et al., 2022]. Among these studies, those using the follow-the-regularized-leader framework [McMahan, 2011] are particularly relevant to our work. In an analysis of algorithms in this category, we show regret bounds that depend on output distributions, and we apply the self-bounding technique to derive BOBW regret bounds. In applying this approach to partial feedback problems including MAB, it has been shown that regularization based on the Tsallis entropy [Zimmert and Seldin, 2021, Zimmert et al., 2019] or the logarithmic barrier [Wei and Luo, 2018, Ito, 2021c, Ito et al., 2022] is useful. By way of contrast, our study employs regularization based on the Shannon entropy and demonstrates for the first time that the self-bounding technique can be applied even with such regularization.

This study includes the regret bounds for stochastic environments with adversarial corruptions [Lykouris et al., 2018, Gupta et al., 2019, Amir et al., 2020], which is an intermediate setting between stochastic and adversarial settings. Zimmert and Seldin [2021] have demonstrated that the self-bounding technique is also useful in deriving regret bounds for corrupted stochastic environments. Typically, when the self-bounding technique yields a regret bound of $O(\mathcal{R})$ for stochastic environments, it also yields a bound of $O(\mathcal{R} + \sqrt{C\mathcal{R}})$ for corrupted stochastic environments, where $C$ represents the amount of corruption. Examples of such results can be found in the literature, e.g., that by Zimmert and Seldin [2021], Erez and Koren [2021], and Ito [2021b]. Our study follows the same strategy as these studies to obtain regret bounds for corrupted stochastic environments.

The problem of online learning with feedback graphs was formulated by Mannor and Shamir [2011], and Alon et al. [2015] have provided a full characterization of minimax regret w.r.t. this problem. Whereas these studies have considered adversarial models, Caron et al. [2012] have considered stochastic settings and proposed an algorithm with an $O(\ln T)$-regret bound. In addition to these,

there can be found studies on such various extensions as models with (uninformed) time-varying feedback graphs [Cohen et al., 2016, Alon et al., 2017], stochastic feedback graphs [Kocák et al., 2016, Ghari and Shen, 2022, Esposito et al., 2022], non-stationary environments [Lu et al., 2021a], and corrupted environments [Lu et al., 2021b] as well as such improved algorithms as those with problem-dependent regret bounds [Hu et al., 2020].

# 3 Problem setting and known results

Let $G = (V, E)$ be a directed graph with $V = [K] = \{1, 2, \ldots, K\}$ and $E \subseteq V \times V$, which we refer to as a *feedback graph*. For each $i \in V$, we denote the in-neighborhood and the out-neighborhood of $i$ in $G$ by $N^{\mathrm{in}}(i)$ and $N^{\mathrm{out}}(i)$, respectively, i.e., $N^{\mathrm{in}}(i) = \{j \in V \mid (j, i) \in E\}$ and $N^{\mathrm{out}}(i) = \{j \in V \mid (i, j) \in E\}$.

Before a game starts, the player is given $G$. For each round $t = 1, 2, \ldots$, the environment selects the loss functions $\ell_t : V \to [0, 1]$, and the player then chooses $I_t \in V$ without knowing $\ell_t$, where the value $\ell_t(i)$ represents the loss for choosing $i \in V$ in the $t$-th round. After that, the player incurs the loss of $\ell_t(I_t)$ and observes $\ell_t(j)$ for all $j \in N^{\mathrm{out}}(I_t)$. Note that the player cannot observe the incurred loss if $I_t \notin N^{\mathrm{out}}(I_t)$. The goal of the player is to minimize the sum of incurred loss. To evaluate performance, we use the regret $R_T$ defined by

$$R_T(i^*) = \mathbf{E}\left[\sum_{t=1}^{T} \ell_t(I_t) - \sum_{t=1}^{T} \ell_t(i^*)\right], \quad R_T = \max_{i^* \in V} R_T(i^*), \quad (1)$$

where the expectation is taken with respect to the randomness of $\ell_t$ and the algorithm's internal randomness. The *minimax regret* $R(G, T)$ is defined as the minimum over all randomized algorithms, of the maximum of $R_T$ over all loss sequences $\{\ell_t\}$. Alon et al. [2015] have shown that the minimax regret can be characterized by the notion of observability:

**Definition 1** ([Alon et al., 2015]). A graph $G$ is *observable* if $N^{\mathrm{in}}(i) \neq \emptyset$ holds for each $i \in V$. A graph $G$ is *strongly observable* if $\{i\} \subseteq N^{\mathrm{in}}(i)$ or $V \setminus \{i\} \subseteq N^{\mathrm{in}}(i)$ holds for each $i \in V$. A graph $G$ is *weakly observable* if it is observable but not strongly observable.

We further define the independence number $\alpha(G)$ and the weak domination number $\delta(G)$ as follows:

**Definition 2.** For a graph $G = (V, E)$, an *independent set* $S \subseteq V$ is a set of vertices such that $u, v \in S, u \neq v \implies (u, v) \notin E$. The *independence number* $\alpha(G)$ of $G$ is the size of its largest independent set. For a graph $G = (V, E)$, a *weakly dominating set* $D \subseteq V$ is a set of vertices such that $\{i \in V \mid i \notin N^{\mathrm{out}}(i)\} \subseteq \bigcup_{i \in D} N^{\mathrm{out}}(i)$. The *weak domination number* $\delta(G)$ of $G$ is the size of its smallest weakly dominating set.

**Remark 2.** The definitions of weakly dominating set and weak domination number in this paper are slightly different from those by Alon et al. [2015]. However, this difference is negligible as the gap between weak domination numbers in our definition and in theirs is at most one. Details are discussed in Appendix D.

The minimax regret can then be characterized as follows:

**Theorem 3** ([Alon et al., 2015]). *Let $G$ be a feedback graph with $|V| \geq 2$. Then, the minimax regret for $T \geq |V|^3$ is (i) $R(G, T) = \tilde{\Theta}(\alpha^{1/2} T^{1/2})$ if $G$ is strongly observable; (ii) $R(G, T) = \tilde{\Theta}(\delta^{1/3} T^{2/3})$ if $G$ is weakly observable; (iii) $R(G, T) = \Theta(T)$ if $G$ is not observable.*

Following this statement by Alon et al. [2015], we assume $T \geq |V|^3 = K^3$ in this paper.

Regret bounds by Erez and Koren [2021] listed in Table 1 depend on the *clique covering number* $\theta(G)$ of the feedback graph. The clique covering number $\theta(G)$ is the minimum value of $N$ such that there exists a *clique cover* $\{V_k\}_{k=1}^{N}$ for $G$ of size $N$. A clique cover is a partition of vertices $V$ such that each $V_k$ is a clique, i.e., $V_k \cap V_{k'} = \emptyset$ for all $k \neq k'$, $\bigcup_{k=1}^{\theta} V_k = V$, and $V_k \times V_k \subseteq E$ holds for any $k$. While there exists an example such that $K = \theta(G) > \alpha(G) = 1$,[1] we always have $\theta(G) \geq \alpha(G)$, that is, the clique covering number is at least the independence number. In fact, for any clique cover $\{V_k\}_{k=1}^{N}$ and any independence set $S \subseteq V$, two distinct elements in $S$ can never be in a single clique $V_k$, which implies that $N \geq |S|$.

---

[1]For example, consider the graph $G = (V, E)$ given by $V = [K]$ and $E = \{(i, j) \in V \times V \mid i \geq j\}$.

In this work, we consider the *adversarial regime with a self-bounding constraint*, a comprehensive regime including stochastic settings, adversarial settings, and adversarially corrupted stochastic settings.

**Definition 3** (adversarial regime with a self-bounding constraint [Zimmert and Seldin, 2021]). Let $\Delta : V \to [0, 1]$ and $C \geq 0$. The environment is in an *adversarial regime with a $(\Delta, C, T)$ self-bounding constraint* if it holds for any algorithm that

$$R_T \geq \mathbf{E}\left[\sum_{t=1}^{T} \Delta(I_t) - C\right]. \tag{2}$$

As has been shown by Zimmert and Seldin [2021], this regime includes (adversarially corrupted) stochastic settings. Indeed, if $\ell_t$ follows a distribution $\mathcal{D}$ independently for $t = 1, 2, \ldots, T$ we have $R_T = \max_{i^* \in V} \mathbf{E}\left[\sum_{t=1}^{T} (\ell_t(I_t) - \ell_t(i^*))\right] = \mathbf{E}\left[\sum_{t=1}^{T} \Delta(I_t)\right]$, where we define $\Delta$ by $\Delta(i) = \mathbf{E}_{\ell \sim \mathcal{D}}[\ell(i)] - \min_{i^* \in V} \mathbf{E}_{\ell \sim \mathcal{D}}[\ell(i^*)]$. This means that the environment is in an adversarial regime with a $(\Delta, 0, T)$ self-bounding constraint. Further, if $\ell_t$ satisfies $\sum_{t=1}^{T} \max_{i \in [N]} |\ell_t(i) - \ell'_t(i)| \leq C$ for some $\ell'_t \sim \mathcal{D}$, the environment is in an adversarial regime with a $(\Delta, C, T)$ self-bounding constraint. Note also that, for any $\Delta : V \to [0, 1]$, the adversarial regime with a $(\Delta, 2T, T)$ self-bounding constraint includes all the adversarial environments since (2) clearly holds when $C = 2T$.

In this paper, we assume that there exists $i^* \in V$ such that $\Delta(i^*) = 0$ and that $\Delta_{\min} := \min_{i \in V \setminus \{i^*\}} \Delta_i > 0$. This implies that the optimal arm $i^*$ is assumed to be unique. Similar assumptions were also made in previous works using the self-bounding technique [Gaillard et al., 2014, Luo and Schapire, 2015, Wei and Luo, 2018, Zimmert and Seldin, 2021, Erez and Koren, 2021].

## 4 Preliminary

The proposed algorithms are based on the follow-the-regularized-leader approach. In this approach, we define a probability distribution $p_t$ over $V$ as follows:

$$q_t \in \underset{p \in \mathcal{P}(V)}{\arg\min} \left\{ \sum_{s=1}^{t-1} \left\langle \hat{\ell}_s, p \right\rangle + \psi_t(p) \right\}, \quad p_t = (1 - \gamma_t)q_t + \gamma_t \mu_U, \tag{3}$$

where $\mathcal{P}(V) = \{p : V \to [0, 1] \mid \sum_{i \in V} p(i) = 1\}$ expresses the set of all probability distributions over $V$, $\hat{\ell}_s$ is an unbiased estimator for $\ell_s$, $\langle \ell, p \rangle = \sum_{i \in V} \ell(i)p(i)$ represents the inner product, $\psi_t : \mathcal{P} \to \mathbb{R}$ is a convex regularizer function, $\gamma_t \in [0, 0.5]$ is a parameter, and $\mu_U$ is the uniform distribution over a nonempty subset $U \subseteq V$, i.e., $\mu_U(i) = 1/|U|$ for $i \in U$ and $\mu_U(i) = 0$ for $i \in V \setminus U$. After computing $p_t$ defined by (3), we choose $I_t$ following $p_t$ so that $\Pr[I_t = i | p_t] = p_t(i)$. We then observe $\ell_t(j)$ for each $j \in N^{\text{out}}(I_t)$. Based on these observations, we set the unbiased estimator $\hat{\ell}_t : V \to \mathbb{R}$ by

$$\hat{\ell}_t(i) = \frac{\ell_t(i)}{P_t(i)} \mathbf{1}[i \in N^{\text{out}}(I_t)], \quad P_t(i) = \sum_{j \in N^{\text{in}}(i)} p_t(j). \tag{4}$$

Let $D_t$ denote the *Bregman divergence* with respect to $\psi_t$, i.e.,

$$D_t(p, q) = \psi_t(p) - \psi_t(q) - \langle \nabla \psi_t(q), p - q \rangle. \tag{5}$$

We then have the following regret bounds:

**Lemma 1.** *If $I_t$ is chosen by the above procedure, the regret is bounded by*

$$R_T \leq \mathbf{E}\left[\sum_{t=1}^{T} \left(\gamma_t + \left\langle \hat{\ell}_t, q_t - q_{t+1}\right\rangle - D_t(q_{t+1}, q_t) + \psi_t(q_{t+1}) - \psi_{t+1}(q_{t+1})\right)\right]$$
$$+ \psi_{T+1}(\mu_{i^*}) - \psi_1(q_1), \tag{6}$$

*where $\mu_{i^*}(i) = 1$ if $i = i^*$ and $\mu_{i^*}(i) = 0$ for $i \in V \setminus \{i^*\}$.*

This lemma can be shown by the standard analysis technique for FTRL, e.g., given in Exercise 28.12 of the book by Lattimore and Szepesvári [2020], combined with the fact that $\hat{\ell}_t$ defined by (4) is an unbiased estimator of $\ell_t$. All omitted proofs will be given in the appendix.

We also introduce the following parameters $Q(i^*)$ and $Q$, which will be used when applying self-bounding technique:

$$Q(i^*) = \sum_{t=1}^{T}(1 - q_t(i^*)), \quad \bar{Q}(i^*) = \mathbf{E}\left[Q(i^*)\right], \quad \bar{Q} = \min_{i^* \in V} \bar{Q}(i^*). \tag{7}$$

We note that these values are clearly bounded as $0 \leq \bar{Q} \leq \bar{Q}(i^*) \leq T$ for any $i^* \in V$. In an adversarial regime with a self-bounding constraint, the regret can be bounded from below, as follows:

**Lemma 2.** *In an adversarial regime with a self-bounding constraint given in Definition 3, the regret is bounded as $R_T \geq \frac{\Delta_{\min}}{2}\bar{Q} - C$.*

This lemma will be used to show poly-logarithmic regret in adversarial regime with a self-bounding constraint.

## 5  Strongly observable case

This section provides an algorithm achieving regret bounds in Theorem 1. We set $U = V$ and define $\psi_t$ using the Shannon entropy $H(p)$ as follows:

$$\psi_t(p) = -\beta_t H(p), \quad \text{where} \quad H(p) = \sum_{i \in V} p(i) \ln \frac{1}{p(i)}, \tag{8}$$

where $\beta_t > 0$ will be defined later. If we choose $\gamma_t = \min\left\{\left(\frac{1}{\alpha T}\right)^{1/2}, \frac{1}{2}\right\}$ and $\beta_t = \frac{1}{2\gamma_t}$ for all $t$, the FTRL algorithm (3) with (8) coincides with the Exp3.G algorithm with the parameter setting given in Theorem 2 (i) by Alon et al. [2015]. As shown by them, this round-independent parameter setting leads to a regret bound of $R_T = O(\alpha^{1/2}T^{1/2}\ln(KT))$.

In this work, we modify the update rule of $\beta_t$ and $\gamma_t$ as follows: We set $\beta_1 = c_1 \geq 1$ and update $\beta_t$ and $\gamma_t$ by

$$\beta_{t+1} = \beta_t + \frac{c_1}{\sqrt{1 + (\ln K)^{-1}\sum_{s=1}^{t} a_s}}, \quad \gamma_t = \frac{1}{2\beta_t}, \tag{9}$$

where $a_s$ is defined by $a_s = H(q_s)$. In the following, we will show the following regret bounds:

**Theorem 4.** *If the feedback graph $G$ is strongly observable and has the independent number $\alpha = \alpha(G)$, the FTRL algorithm (3) with $U = V$ and $\psi_t$ defined by (8) and (9) enjoys a regret bound of*

$$R_T \leq \hat{c} \cdot \max\left\{\bar{Q}^{1/2}, 1\right\}, \quad \text{where } \hat{c} = O\left(\left(\frac{\alpha \ln T \cdot \ln(c_1 KT)}{c_1\sqrt{\ln K}} + c_1\sqrt{\ln K}\right)\sqrt{\ln(KT)}\right). \tag{10}$$

*Consequently, we have $R_T = O\left(\hat{c}\sqrt{T}\right)$ in the adversarial regime and $R_T = O\left(\frac{\hat{c}^2}{\Delta_{\min}} + \sqrt{\frac{C\hat{c}^2}{\Delta_{\min}}}\right)$ in adversarial regimes with self-bounding constraints.*

When we set $c_1 = \Theta\left(\sqrt{\frac{\alpha \ln T \cdot \ln(KT)}{\ln K}}\right)$, $\hat{c}$ in this theorem is at most $O\left(\sqrt{\alpha \ln T \cdot (\ln(KT))^2}\right)$, which leads to the regret bounds in Theorem 1. In the rest of this section, we provide proof for Theorem 4.

Let us start with the following lemma:

**Lemma 3.** *If $\psi_t$ is given by (8) with $\beta_t \geq 1$ and $\gamma_t \geq 1/(2\beta_t)$, the regret for the FTRL algorithm (3) with $U = V$ is bounded as*

$$R_T \leq \mathbf{E}\left[\sum_{t=1}^{T}\left(\gamma_t + \frac{2}{\beta_t}\left(1 + 4\alpha \ln \frac{K^2}{4\gamma_t}\right) + (\beta_{t+1} - \beta_t)a_{t+1}\right)\right] + \beta_1 \ln K, \tag{11}$$

*where $a_t = H(q_t)$ is the value of the Shannon entropy for $q_t$.*

This lemma follows from Lemma 1 and the technique used in the proof of Alon et al. [2015, Theorem 2]. We note that $0 \le a_t \le \ln K$ and $a_1 = \ln K$. From Lemma 3 and the update rules of parameters given by (9), we obtain the following entropy-dependent regret bound:

**Proposition 1.** *Suppose* (11) *holds. If $\beta_t$ and $\gamma_t$ are given by* (9), $R_T \le \tilde{c}\,\mathbf{E}\left[\sqrt{\sum_{t=1}^{T} a_t}\right]$, *where*

$a_t = H(q_t)$ *and* $\tilde{c} = O\left(\frac{\alpha \ln T \cdot \ln(c_1 KT)}{c_1 \sqrt{\ln K}} + c_1 \sqrt{\ln K}\right).$

*Proof.* We will show the following two inequalities:

$$\sum_{t=1}^{T}\left(\gamma_t + \frac{2}{\beta_t}\left(1 + 4\alpha \ln \frac{K^2}{4\gamma_t}\right)\right) = O\left(\frac{\alpha \ln T \cdot \ln(c_1 K^2 T)}{c_1 \sqrt{\ln K}}\sqrt{\sum_{t=1}^{T} a_t}\right), \tag{12}$$

$$\sum_{t=1}^{T}(\beta_{t+1} - \beta_t)a_{t+1} = O\left(c_1\sqrt{\ln K}\sqrt{\sum_{t=1}^{T} a_t}\right). \tag{13}$$

Let us first show (12). From the definition of $\gamma_t$ given in (9), we have [LHS of (12)] $\le \sum_{t=1}^{T}\frac{1}{\beta_t}\left(3 + 8\alpha \ln \frac{c_1 K^2 t}{2}\right) \le \left(3 + 8\alpha \ln \frac{c_1 K^2 T}{2}\right)\sum_{t=1}^{T}\frac{1}{\beta_t}$. From the definition of $\beta_t$ given by (9), $\beta_t$ is bounded as $\beta_t = c_1 + \sum_{u=1}^{t-1}\frac{c_1}{\sqrt{1+(\ln K)^{-1}\sum_{s=1}^{u} a_s}} \ge \frac{c_1 t}{\sqrt{1+(\ln K)^{-1}\sum_{s=1}^{t} a_s}}$. We hence have $\sum_{t=1}^{T}\frac{1}{\beta_t} \le \sum_{t=1}^{T}\frac{1}{c_1 t}\sqrt{1+(\ln K)^{-1}\sum_{s=1}^{t} a_s} \le \frac{1+\ln T}{c_1}\sqrt{1+(\ln K)^{-1}\sum_{t=1}^{T} a_t} \le O\left(\frac{\ln T}{c_1\sqrt{\ln K}}\sqrt{\sum_{t=1}^{T} a_t}\right)$, where the last inequality follows from $a_1 = \ln K$. Combining the above inequalities, we obtain (12).

Let us next show (13). From (9), we have [LHS of (13)] $= \sum_{t=1}^{T}\frac{c_1}{\sqrt{1+(\ln K)^{-1}\sum_{s=1}^{t} a_s}} \cdot a_{t+1} = 2c_1\sqrt{\ln K}\sum_{t=1}^{T}\frac{a_{t+1}}{\sqrt{\ln K + \sum_{s=1}^{t} a_s} + \sqrt{\ln K + \sum_{s=1}^{t} a_s}} \le 2c_1\sqrt{\ln K}\sum_{t=1}^{T}\frac{a_{t+1}}{\sqrt{\sum_{s=1}^{t+1} a_s} + \sqrt{\sum_{s=1}^{t} a_s}} = 2c_1\sqrt{\ln K}\sum_{t=1}^{T}\left(\sqrt{\sum_{s=1}^{t+1} a_s} - \sqrt{\sum_{s=1}^{t} a_s}\right) = 2c_1\sqrt{\ln K}\left(\sqrt{\sum_{s=1}^{T+1} a_s} - \sqrt{a_1}\right) \le 2c_1\sqrt{\ln K}\sqrt{\sum_{t=1}^{T} a_t}$, where inequalities follow from $a_t \le a_1 = \ln K$. This proves (13).

Inequalities (12) and (13) combined with (11) lead to the regret bound in Proposition 1. $\qquad\square$

In addition, $\sum_{t=1}^{T} a_t = \sum_{t=1}^{T} H(q_t)$ is bounded with $Q(i^*)$ defined in (7), as follows:

**Lemma 4.** *Suppose $a_t = H(q_t)$. For any $i^* \in V$, we have $\sum_{t=1}^{T} a_t \le Q(i^*)\ln \frac{\mathrm{e}KT}{Q(i^*)}$.*

We are now ready to prove Theorem 4.
*Proof of Theorem 4.* From Lemma 4, if $Q(i^*) \le \mathrm{e}$, we have $\sum_{t=1}^{T} a_t \le \mathrm{e}\ln(KT)$ and otherwise, we have $\sum_{t=1}^{T} a_t \le Q(i^*)\ln(KT)$. Hence, we have $\sum_{t=1}^{T} a_t \le \ln(KT) \cdot \max\{\mathrm{e}, Q(i^*)\}$. Combining this with Proposition 1, we obtain (10). Since $\bar{Q} \le T$, we have $R_T \le \hat{c}\sqrt{T}$ in adversarial regimes.

We next show $R_T = O\left(\frac{\hat{c}^2}{\Delta_{\min}} + \sqrt{\frac{C\hat{c}^2}{\Delta_{\min}}}\right)$. From Lemma 2, (2) implies if the environment satisfies a $(\Delta, C, T)$ self-bounding constraint (2), we have $R_T \ge \frac{\Delta_{\min}}{2}\bar{Q} - C$. Combining this with Proposition 1 and Lemma 4, it holds for any $\lambda > 0$ that

$$R_T = (1+\lambda)R_T - \lambda R_T \le (1+\lambda)\tilde{c}\sqrt{\bar{Q}\ln(KT)} - \frac{\lambda\Delta_{\min}}{2}\bar{Q} + \lambda C$$

$$\le \frac{((1+\lambda)\tilde{c})^2 \ln(KT)}{2\lambda\Delta_{\min}} + \lambda C = \frac{\tilde{c}^2\ln(KT)}{\Delta_{\min}} + \frac{1}{2\lambda}\frac{\tilde{c}^2\ln(KT)}{\Delta_{\min}} + \frac{\lambda}{2}\left(\frac{\tilde{c}^2\ln(KT)}{\Delta_{\min}} + 2C\right),$$

where the first inequality follows from Proposition 1, Lemma 4, the condition of $Q(i^*) \ge \mathrm{e}$, and (23). The second inequality follows from $a\sqrt{x} - \frac{b}{2}x = \frac{a^2}{2b} - \frac{1}{2}\left(\frac{a}{\sqrt{b}} - \sqrt{bx}\right) \le \frac{a^2}{2b}$ which

holds for any $a, b, x \geq 0$. By choosing $\lambda = \sqrt{\frac{\tilde{c}^2 \ln(KT)}{\Delta_{\min}} / \left( \frac{\tilde{c}^2 \ln(KT)}{\Delta_{\min}} + 2C \right)}$, we obtain $R_T = O\left( \frac{\hat{c}^2}{\Delta_{\min}} + \sqrt{\frac{C\hat{c}^2}{\Delta_{\min}}} \right)$. $\qquad\square$

## 6 Weakly observable case

This section provides an algorithm achieving regret bounds in Theorem 2. Let $D$ be a weakly dominating set, defined in Definition 2, and let $V_1 = \bigcup_{i \in D} N^{\text{out}}(i)$, $V_2 = V \setminus V_1$. We consider here the FTRL approach given by (3) with $U = D$ and regularizer functions defined as

$$\psi_t(p) = \beta_t \sum_{i \in V_1} h(p(i)) + \sum_{i \in V_2} \sqrt{t} g(p(i)),$$

$$\text{where} \quad h(x) = x \ln x + (1-x) \ln(1-x), \quad g(x) = -2\sqrt{x} - 2\sqrt{1-x}. \tag{14}$$

The regularization with $h(x)$ for $V_1$ is a variant of Shannon-entropy regularization, which can be considered as a modification of the approach of the Exp3.G by Alon et al. [2015]. The remaining part defined with $g(x)$ for $V_2$ is a modification of the approach used in the Tsallis-INF algorithm by Zimmert and Seldin [2021], which is a BOBW algorithm for MAB problems. Intuitively, approaches for MAB work well for vertices in $V_2$ as they have self-loops, i.e., choosing actions in $V_2$ admits bandit feedback.

Let us define parameters $\gamma_t$ and $\beta_t$ by $\beta_1 = \max\{c_2, 8|D|\}$ and

$$\gamma_t' = \frac{1}{4} \frac{c_1 b_t}{c_1 + \left( \sum_{s=1}^t b_s \right)^{1/3}}, \quad \beta_{t+1} = \beta_t + \frac{c_2 b_t}{\gamma_t' \left( c_1 + \sum_{s=1}^{t-1} \frac{b_s a_{s+1}}{\gamma_s'} \right)^{1/2}}, \quad \gamma_t = \gamma_t' + \frac{2|D|}{\beta_t}, \tag{15}$$

where $c_1, c_2 > 0$ are input parameters such that $c_1 \geq 2 \ln K$ and with $\{a_t\}$ and $\{b_t\}$ are defined by

$$a_t = -\sum_{i \in V_1} h(q_t(i)), \quad b_t = \sum_{i \in V_1} q_t(i)(1 - q_t(i)). \tag{16}$$

Note that $a_t$ and $\hat{c}$ used in this Section 6 are different from those defined in Section 5. We then have the following regret bounds:

**Theorem 5.** *If the feedback graph $G$ is weakly observable, the FTRL algorithm* (3) *with $U = D$ and $\psi_t$ defined by* (14) *and* (15) *enjoys a regret bound of*

$$R_T \leq \hat{c} \cdot \max\left\{ \bar{Q}^{2/3}, c_1^2 \right\} + O\left( (|V_2| \ln T \cdot \bar{Q})^{1/2} \right) \quad \text{where}$$

$$\hat{c} = O\left( c_1 + \frac{1}{\sqrt{c_1}} \left( \frac{|D| \ln T}{c_2} + c_2 \right) \sqrt{\ln(KT)} \right). \tag{17}$$

*Consequently, if $T \geq K^3$, we have $R_T = O\left( \hat{c} T^{2/3} \right)$ in the adversarial regime and*

$$R_T = O\left( \frac{\hat{c}^3}{\Delta_{\min}^2} + \left( \frac{C^2 \hat{c}^3}{\Delta_{\min}^2} \right)^{1/3} + \frac{|V_2| \ln T}{\Delta_{\min}} + \sqrt{\frac{C|V_2| \ln T}{\Delta_{\min}}} \right) \tag{18}$$

*in adversarial regimes with self-bounding constraints.*

We obtain $\hat{c} = O\left( (|D| \ln T \cdot \ln(KT))^{1/3} \right)$ by setting $c_1 = \Theta\left( (|D| \ln T \cdot \ln(KT))^{1/3} \right)$ and $c_2 = \Theta\left( \sqrt{|D| \ln T} \right)$. By using a weakly dominating set $D$ such that $|D| = O(\delta(G))$, we obtain the regret bounds in Theorem 2. The remainder of this section is dedicated to the proof of Theorem 5.

We start with the following regret bound:

**Lemma 5.** *If $\psi_t$ is given by* (14) *and if $\gamma_t \geq \frac{2|D|}{\beta_t}$, we have $R_T \leq R_T^{(1)} + R_T^{(2)} + a_1\beta_1$, where*

$$R_T^{(1)} = O\left(\mathbf{E}\left[\sum_{t=1}^{T}\left(\gamma_t + \frac{|D|b_t}{\gamma_t\beta_t} + (\beta_{t+1} - \beta_t)a_{t+1}\right)\right]\right), \tag{19}$$

$$R_T^{(2)} = O\left(\mathbf{E}\left[\sum_{t=1}^{T}\frac{1}{\sqrt{t}}\sum_{i \in V_2}\sqrt{q_t(i)(1 - q_t(i))}\right]\right), \tag{20}$$

*with $\{a_t\}$ and $\{b_t\}$ defined by* (16).

When showing (19) and (20), we use techniques used in the proofs of Alon et al. [2015, Theorem 2] and of Zimmert and Seldin [2021, Lemma 11]. We then have the following bound:

**Proposition 2.** *If $\gamma_t$ and $\beta_t$ are given by* (15), *$R_T^{(1)}$ satisfying* (19) *is bounded as*

$$R_T^{(1)} = O\left(\mathbf{E}\left[c_1 B_T^{2/3} + \tilde{c}\sqrt{c_1^2 + (\ln K + A_T)\left(c_1 + B_T^{1/3}\right)}\right]\right), \tag{21}$$

*where $A_T = \sum_{t=1}^{T} a_t$, $B_T = \sum_{t=1}^{T} b_t$ and $\tilde{c} = O\left(\frac{1}{\sqrt{c_1}}\left(\frac{|D|\ln T}{c_2} + c_2\right)\right)$.*

Values of $A_T$ and $B_T$ in this proposition can be bounded with $Q(i^*)$ defined in (7), as follows:

**Lemma 6.** *$A_T$ and $B_T$ defined in Proposition 2 satisfy $A_T \leq 2Q(i^*)\ln\frac{eKT}{Q(i^*)}$ and $B_T \leq 2Q(i^*)$.*

Further, $R_T^{(2)}$ in Lemma 5 can be bounded with $\bar{Q}$ as follows:

**Lemma 7.** *$R_T^{(2)}$ satisfying* (20) *is bounded as $R_T^{(2)} = O\left(\sqrt{|V_2|\ln T \cdot \bar{Q}}\right)$.*

*Proof of Theorem 5.* From Proposition 2 and Lemma 6, if $\bar{Q} \geq c_1^3$, we have

$$R_T^{(1)} = O\left(\mathbf{E}\left[c_1 Q(i^*)^{2/3} + \tilde{c}\sqrt{Q(i^*)\ln(KT)Q(i^*)^{1/3}}\right]\right) \leq O\left(\left(c_1 + \tilde{c}\sqrt{\ln(KT)}\right)\bar{Q}^{2/3}\right),$$

where the inequality follows from Jensen's inequality. Hence, there exists $\hat{c}$ such that $R_T^{(1)} \leq \hat{c} \cdot \bar{Q}^{2/3}$ and $\hat{c} = O\left(c_1 + \tilde{c}\sqrt{\ln(KT)}\right)$. Combining this with Lemma 7, we obtain (17). As we have $\bar{Q} \leq T$, in adversarial regimes with $T \geq K^3$, it follows from (17) that $R_T = O\left(\hat{c} \cdot \max\{T^{2/3}, c_1^2\} + (K\ln T \cdot T)^{1/2}\right) = O\left(\hat{c} \cdot T^{2/3}\right)$, where the second equality follows from the $T \geq K^3$. Let us next show (18). From (17) and Lemma 2, for any $\lambda \in (0, 1]$, we have

$$R_T = (1 + \lambda)R_T - \lambda R_T = O\left((1 + \lambda)\hat{c} \cdot \bar{Q}^{2/3} + (1 + \lambda)(|V_2|\ln T \cdot \bar{Q})^{1/2} - \lambda\Delta_{\min}\bar{Q} + \lambda C\right).$$

By an argument similar to the proof of Theorem 4, we have $(1 + \lambda)(|V_2|\ln T \cdot \bar{Q})^{1/2} - \lambda\Delta_{\min}\bar{Q} = O\left(\left(1 + \frac{1}{\lambda}\right)\frac{|V_2|\ln T}{\Delta_{\min}}\right)$. We also have $(1 + \lambda)\hat{c} \cdot \bar{Q}^{2/3} - \lambda\Delta_{\min}\bar{Q} = \left(\frac{(1+\lambda)^3\hat{c}^3}{\lambda^2\Delta_{\min}^2}\right)^{1/3}\left(\lambda\Delta_{\min}\bar{Q}\right)^{2/3} - \lambda\Delta_{\min}\bar{Q} = O\left(\frac{(1+\lambda)^3\hat{c}^3}{\lambda^2\Delta_{\min}^2}\right) = O\left(\left(1 + \frac{1}{\lambda^2}\right)\frac{\hat{c}^3}{\Delta_{\min}^2}\right)$, where the second equality follows from $x^{1/3}y^{2/3} \leq \frac{1}{3}x + \frac{2}{3}y$ that holds for any $x, y \geq 0$. Combining these inequalities, we obtain $R_T = O\left(\left(1 + \frac{1}{\lambda^2}\right)\frac{\hat{c}^3}{\Delta_{\min}^2} + \left(1 + \frac{1}{\lambda}\right)\frac{|V_2|\ln T}{\Delta_{\min}} + \lambda C\right)$. By choosing $\lambda$ that minimizes the RHS, we obtain (18). $\qquad\square$

## Acknowledgment

TT was supported by JST, ACT-X Grant Number JPMJAX210E, Japan and JSPS, KAKENHI Grant Number JP21J21272, Japan. JH was supported by JSPS, KAKENHI Grant Number JP21K11747, Japan.

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
