# A Omitted proofs

## A.1 Proof of Lemma 1

*Proof.* From the definition of the algorithm, we have

$$R_T(i^*) = \mathbf{E}\left[\sum_{t=1}^{T} \ell_t(I_t) - \sum_{t=1}^{T} \ell_t(i^*)\right] = \mathbf{E}\left[\sum_{t=1}^{T} \langle \ell_t, p_t - \mu_{i^*} \rangle\right]$$

$$= \mathbf{E}\left[\sum_{t=1}^{T} \langle \ell_t, q_t - \mu_{i^*} \rangle + \sum_{t=1}^{T} \gamma_t \langle \ell_t, \mu_U - q_t \rangle\right] \leq \mathbf{E}\left[\sum_{t=1}^{T} \langle \ell_t, q_t - \mu_{i^*} \rangle + \sum_{t=1}^{T} \gamma_t\right]$$

$$= \mathbf{E}\left[\sum_{t=1}^{T} \left\langle \hat{\ell}_t, q_t - \mu_{i^*} \right\rangle + \sum_{t=1}^{T} \gamma_t\right], \tag{22}$$

where the second equality follows from $I_t \sim p_t$, the third equality follows from the second part of (3), the first inequality follows from $\langle \ell_t, \mu_U - q_t \rangle \leq \langle \ell_t, \mu_U \rangle \leq 1$, and the last equality follows from the fact that $\hat{\ell}_t$ is an unbiased estimator for $\ell_t$. Further, from Exercise 28.12 of the book by Lattimore and Szepesvári [2020], we have

$$\sum_{t=1}^{T} \left\langle \hat{\ell}_t, q_t - \mu_{i^*} \right\rangle$$

$$\leq \sum_{t=1}^{T} \left( \left\langle \hat{\ell}_t, q_t - q_{t+1} \right\rangle - D_t(q_{t+1}, q_t) + \psi_t(q_{t+1}) - \psi_{t+1}(q_{t+1}) \right) + \psi_{T+1}(\mu_{i^*}) - \psi_1(q_1).$$

Combining this with (22), we obtain (6). □

## A.2 Proof of Lemma 2

*Proof.* Suppose that (2) holds with $\Delta : V \to \mathbb{R}$ such that $\Delta(i) \geq \Delta_{\min}$ for all $i \in [K] \setminus \{i^*\}$. The regret is then bounded as

$$R_T \geq \mathbf{E}\left[\sum_{t=1}^{T} \Delta(I_t)\right] - C = \mathbf{E}\left[\sum_{t=1}^{T} \sum_{i \in V} \Delta(i) p_t(i)\right] - C$$

$$\geq \mathbf{E}\left[\sum_{t=1}^{T} \sum_{i \in V} \Delta(i)(1 - \gamma_t) q_t(i)\right] - C \geq \mathbf{E}\left[\frac{\Delta_{\min}}{2} Q(i^*)\right] - C \geq \frac{\Delta_{\min}}{2} \bar{Q} - C, \tag{23}$$

where the first inequality follows from (2), the first equality follows from $I_t \sim p_t$, the second inequality follows from the definition of $p_t$ given in (3), and the third and last inequalities follow from the assumption of $\gamma_t \leq \frac{1}{2}$ and the definitions of $Q(i^*)$ and $\bar{Q}$ given in (7). This completes the proof of Lemma 2. □

## A.3 Proof of Lemma 3

We use the following lemma to analyze the right-hand sided of (6).

**Lemma 8.** *If $\psi_t$ is given by* (8)*, it holds for any $\ell : V \to \mathbb{R}$ and $p, q \in \mathcal{P}(V)$ that*

$$\langle \ell, p - q \rangle - D_t(q, p) \leq \beta_t \sum_{i \in V} p(i) \xi\left(\frac{\ell(i)}{\beta_t}\right), \quad where \quad \xi(x) = \exp(-x) + x - 1. \tag{24}$$

*Proof.* The derivative of the LHS of (24) w.r.t. $q(i)$ is expressed as

$$\frac{\partial}{\partial q(i)} \left( \langle \ell, p - q \rangle - D_t(q, p) \right) = -\ell(i) - \beta_t \left( \ln q(i) - \ln p(i) \right). \tag{25}$$

As the LHS of (24) is concave in $q$, its maximum subject to $q : V \to \mathbb{R}_{>0}$ is attained when the values of (25) are equal to zero, i.e., $q(i) = q^*(i) := p(i) \exp\left(-\frac{\ell(i)}{\beta_t}\right)$. Hence, we have

$$
\begin{aligned}
\langle \ell, p - q \rangle - D_t(q, p) &\leq \langle \ell, p - q^* \rangle - D_t(q^*, p) \\
&= \sum_{i \in V} \left( \ell(i)(p(i) - q^*(i)) - \beta_t \left( q^*(i) \ln q^*(i) - p(i) \ln p(i) - (\ln p(i) + 1)(q^*(i) - p(i)) \right) \right) \\
&= \sum_{i \in V} \left( \ell(i)p(i) - \beta_t \left( q^*(i) \ln p(i) - p(i) \ln p(i) - (\ln p(i) + 1)(q^*(i) - p(i)) \right) \right) \\
&= \sum_{i \in V} \left( \ell(i)p(i) + \beta_t \left( (q^*(i) - p(i)) \right) \right) = \beta_t \sum_{i \in V} p(i) \left( \exp\left(-\frac{\ell(i)}{\beta_t}\right) + \frac{\ell(i)}{\beta_t} - 1 \right) \\
&= \beta_t \sum_{i \in V} p(i) \xi\left(\frac{\ell(i)}{\beta_t}\right),
\end{aligned}
$$

where the first equality follows from the definition of the Bregman divergence and (8), the second equality follows from $\ln q^*(i) = \ln p(i) - \frac{\ell(i)}{\beta_t}$, and the fourth inequality follows from $q^*(i) = p(i) \exp\left(-\frac{\ell(i)}{\beta_t}\right)$. This complete the proof of Lemma 8. $\square$

Note that as we have $\exp(-x) \leq 1 - x + x^2$ for any $x \geq -1$, the function $\xi$ defined in (24) satisfies $\xi(x) \leq x^2$ for any $x \geq -1$. Hence, Lemma 8 implies that $\langle \ell, p - q \rangle - D_t(q, p) \leq \beta_t \sum_{i \in V} p(i) \xi\left(\frac{\ell(i)}{\beta_t}\right) \leq \frac{1}{\beta_t} \sum_{i \in V} p(i)\ell(i)^2$ holds for any $\ell : V \to [-\beta_t, \infty)$.

Denote $S = \{i \in V \mid i \notin N^{\mathrm{in}}(i)\}$. From Lemma 8 and the argument by Alon et al. [2015, Lemma 4, Theorem 2], we have

$$
\begin{aligned}
\mathbf{E}\left[ \left\langle \hat{\ell}_t, q_t - q_{t+1} \right\rangle - D_t(q_{t+1}, q_t) \right] &= \mathbf{E}\left[ \left\langle \hat{\ell}_t - \bar{\ell}_t \cdot \mathbf{1}, q_t - q_{t+1} \right\rangle - D_t(q_{t+1}, q_t) \right] \\
&\leq \beta_t \sum_{i \in V} q_t(i) \xi\left( \frac{\hat{\ell}_t(i) - \bar{\ell}_t}{\beta_t} \right) \leq \frac{1}{\beta_t} \left( \sum_{i \in S} q_t(i)(1 - q_t(i))\hat{\ell}_t(i)^2 + \sum_{i \in V \setminus S} q_t(i)\hat{\ell}_t(i)^2 \right), \quad (26)
\end{aligned}
$$

where $\bar{\ell}_t$ is defined in a way similar to by Alon et al. [2015, Lemma 4], the first inequality follows from Lemma 8 and the last inequality follows from the definition of $\bar{\ell}_t$ and the inequality $\xi(x) \leq x^2$ that holds for $x \geq -1$. The first term of the right-hand side of (26) can be bounded as

$$
\begin{aligned}
\mathbf{E}\left[ \sum_{i \in S} q_t(i)(1 - q_t(i))\hat{\ell}_t(i)^2 \right] &= \mathbf{E}\left[ \sum_{i \in S} q_t(i)(1 - q_t(i)) \frac{\ell_t(i)^2 \mathbf{1}[i \in N^{\mathrm{out}}(I_t)]}{P_t(i)^2} \right] \\
&= \mathbf{E}\left[ \sum_{i \in S} q_t(i)(1 - q_t(i)) \frac{\ell_t(i)^2}{P_t(i)} \right] \leq \mathbf{E}\left[ \sum_{i \in S} q_t(i) \frac{1 - q_t(i)}{P_t(i)} \right] \\
&= \mathbf{E}\left[ \sum_{i \in S} q_t(i) \frac{1 - q_t(i)}{1 - p_t(i)} \right] \leq \mathbf{E}\left[ 2 \sum_{i \in S} q_t(i) \right] \leq 2, \quad (27)
\end{aligned}
$$

where the first equality follows from (4), the third equality follows from the assumption of strong observability implying that $N^{\mathrm{in}}(i) = [K] \setminus \{i\}$ for all $i \in S$, and the second inequality follows from the second part of (3) and the assumption of $\gamma_t \in [0, 0.5]$. The second term of the right-hand side of (26) is bounded as

$$
\sum_{i \in V \setminus S} q_t(i)\hat{\ell}_t(i)^2 \leq \mathbf{E}\left[ \sum_{i \in V \setminus S} q_t(i) \frac{1}{P_t(i)} \right] \leq 2\mathbf{E}\left[ \sum_{i \in V \setminus S} p_t(i) \frac{1}{P_t(i)} \right] \leq 8\alpha(G) \ln \frac{K^2}{4\gamma_t}, \quad (28)
$$

where the second inequality follows from the second part of (3) and the assumption of $\gamma_t \in [0, 0.5]$, and the last inequality follows from Lemma 5 by Alon et al. [2015].

Combining (26), (27) and (28), we obtain

$$\mathbf{E}\left[\left\langle \hat{\ell}_t, q_t - q_{t+1} \right\rangle - D_t(q_{t+1}, q_t)\right] \le \frac{2}{\beta_t}\left(1 + 4\alpha(G)\ln\frac{K^2}{4\gamma_t}\right). \tag{29}$$

In addition, from the definition of $\psi_t$ in (8), we have

$$\sum_{t=1}^{T}(\psi_t(q_{t+1}) - \psi_{t+1}(q_{t+1})) + \psi_{T+1}(\mu_{i^*}) - \psi_1(q_1)$$

$$= \sum_{t=1}^{T}(\beta_{t+1} - \beta_t)H(q_{t+1}) - \beta_{T+1}H(\mu_{i^*}) + \beta_1 H(q_1)$$

$$\le \sum_{t=1}^{T}(\beta_{t+1} - \beta_t)H(q_{t+1}) + \beta_1 \ln K.$$

By combining this with (29) and Lemma 1, we obtain (11).

### A.4 Proof of Lemma 4

*Proof.* For any $p \in \mathcal{P}(V)$, and for any $i^* \in V$, we have

$$H(p) = \sum_{i\in V} p(i)\ln\frac{1}{p(i)} = \sum_{i\in V\setminus\{i^*\}} p(i)\ln\frac{1}{p(i)} + p(i^*)\ln\left(1 + \frac{1 - p(i^*)}{p(i^*)}\right)$$

$$\le (K-1)\cdot\frac{\sum_{i\in V\setminus\{i^*\}}p(i)}{K-1}\ln\frac{K-1}{\sum_{i\in V\setminus\{i^*\}}p(i)} + p(i^*)\frac{1 - p(i^*)}{p(i^*)}$$

$$= (1 - p(i^*))\left(\ln\frac{K-1}{1 - p(i^*)} + 1\right), \tag{30}$$

where the inequality follows from Jensen's inequality and $\ln(1 + x) \le x$ that holds for any $x \ge 0$ and the last equality follows from $\sum_{i\in V}p(i) = 1$. Using this, we have

$$\sum_{t=1}^{T}a_t = \sum_{t=1}^{T}H(q_t) \le \sum_{t=1}^{T}(1 - q_t(i^*))\left(\ln\frac{K-1}{1 - q_t(i^*)} + 1\right)$$

$$\le Q(i^*)\left(\ln\frac{(K-1)T}{Q(i^*)} + 1\right) \le Q(i^*)\left(\ln\frac{eKT}{Q(i^*)}\right),$$

where the second inequality follows from Jensen's inequality with the definition $Q(i^*) = \sum_{t=1}^{T}(1 - q_t(i^*))$. $\square$

### A.5 Proof of Lemma 5

We use the following lemma to analyze the right-hand sided of (6).

**Lemma 9.** *If $\psi_t$ is given by (14), it holds for any $\ell : V \to \mathbb{R}$ and $p, q \in \mathcal{P}(V)$ that*

$$\langle \ell, p - q \rangle - D_t(q, p) \le \beta_t \sum_{i\in V_1} \min\left\{p(i)\xi\left(\frac{\ell(i)}{\beta_t}\right), (1 - p(i))\xi\left(-\frac{\ell(i)}{\beta_t}\right)\right\}$$

$$+ \sqrt{t}\sum_{i\in V_2}\min\left\{\sqrt{p(i)}\zeta\left(\frac{\sqrt{p(i)}\ell(i)}{\sqrt{t}}\right), \sqrt{1 - p(i)}\zeta\left(-\frac{\sqrt{1 - p(i)}\ell(i)}{\sqrt{t}}\right)\right\}, \tag{31}$$

$$\text{where}\quad \xi(x) = \exp(-x) + x - 1, \quad \zeta(x) = \frac{x^2}{1 + x}. \tag{32}$$

*Proof.* For any $x, y \in (0, 1)$, we define $d^{(1)}(y, x) \ge 0$ and $d^{(2)}(y, x) \ge 0$ by

$$d^{(1)}(y, x) = y\ln y - x\ln x - (\ln x + 1)(y - x) = y\ln\frac{y}{x} + x - y, \tag{33}$$

$$d^{(2)}(y, x) = -2\sqrt{y} + 2\sqrt{x} + \frac{1}{\sqrt{x}}(y - x) = \frac{1}{\sqrt{x}}\left(\sqrt{y} - \sqrt{x}\right)^2. \tag{34}$$

Note that $d^{(1)}$ and $d^{(2)}$ correspond to Bregman divergences over $(0, 1)$ for $\psi^{(1)}(x) = x \ln x$ and $\psi^{(2)}(x) = -2\sqrt{x}$. If $\psi_t$ is given by (14), the Bregman divergence $D_t(q, p)$ associated with $\psi_t$ is expressed as

$$D_t(q, p) = \beta_t \sum_{i \in V_1} \left( d^{(1)}(q(i), p(i)) + d^{(1)}(1 - q(i), 1 - p(i)) \right)$$
$$+ \sqrt{t} \sum_{i \in V_2} \left( d^{(2)}(q(i), p(i)) + d^{(2)}(1 - q(i), 1 - p(i)) \right).$$

From this, we have

$$\langle \ell, p - q \rangle - D_t(q, p)$$
$$\leq \sum_{i \in V_1} \left( \ell(i)(p(i) - q(i)) - \beta_t(d^{(1)}(q(i), p(i)) + d^{(1)}(1 - q(i), 1 - p(i))) \right)$$
$$+ \sum_{i \in V_2} \left( \ell(i)(p(i) - q(i)) - \sqrt{t}(d^{(2)}(q(i), p(i)) + d^{(2)}(1 - q(i), 1 - p(i))) \right)$$
$$\leq \sum_{i \in V_1} \min \left\{ \ell(i)(p(i) - q(i)) - \beta_t d^{(1)}(q(i), p(i)), \ell(i)(p(i) - q(i)) - \beta_t d^{(1)}(1 - q(i), 1 - p(i)) \right\}$$
$$+ \sum_{i \in V_2} \min \left\{ \ell(i)(p(i) - q(i)) - \sqrt{t}d^{(2)}(q(i), p(i)), \ell(i)(p(i) - q(i)) - \sqrt{t}d^{(2)}(1 - q(i), 1 - p(i)) \right\}.$$
$$(35)$$

By the arguments in the proof of Lemma 8, we have

$$\ell(i)(p(i) - q(i)) - \beta_t d^{(1)}(q(i), p(i)) \leq \beta_t p(i) \xi \left( \frac{\ell(i)}{\beta_t} \right). \tag{36}$$

In a similar way, we can show

$$\ell(i)(p(i) - q(i)) - \beta_t d^{(1)}(1 - q(i), 1 - p(i))$$
$$= -\ell(i)((1 - p(i)) - (1 - q(i))) - \beta_t d^{(1)}(1 - q(i), 1 - p(i)) \leq \beta_t (1 - p(i)) \xi \left( -\frac{\ell(i)}{\beta_t} \right). \tag{37}$$

Let us next evaluate the term $\ell(i)(p(i) - q(i)) - \sqrt{t}d^{(2)}(q(i), p(i))$ in the right-hand side of (35). Denoting $z = \sqrt{q(i)}$, we have

$$\ell(i)(p(i) - q(i)) - \sqrt{t}d^{(2)}(q(i), p(i)) = \ell(i)(p(i) - z^2) - \sqrt{t}\frac{1}{\sqrt{p(i)}} \left( z - \sqrt{p(i)} \right)^2, \tag{38}$$

where the last inequality follows from (34). Hence, its derivative in $z$ can be expressed as

$$-2\ell(i)z - 2\sqrt{t}\frac{1}{\sqrt{p(i)}} \left( z - \sqrt{p(i)} \right) = -2 \left( \ell(i) + \sqrt{\frac{t}{p(i)}} \right) z + 2\sqrt{t}. \tag{39}$$

The value of this expression is equal to zero when $z = z^* := \frac{\sqrt{tp(i)}}{\sqrt{t} + \sqrt{p(i)}\ell(i)}$. As (38) is concave in $z$, its value is maximized when $z = z^*$. Hence, we have

$$\ell(i)(p(i) - q(i)) - \sqrt{t}d^{(2)}(q(i), p(i)) \leq \ell(i)(p(i) - z^{*2}) - \sqrt{t}\frac{1}{\sqrt{p(i)}} \left( z^* - \sqrt{p(i)} \right)^2$$
$$= \left( \sqrt{p(i)} - z^* \right) \left( \ell(i) \left( \sqrt{p(i)} + z^* \right) - \frac{\sqrt{t}}{\sqrt{p(i)}} \left( \sqrt{p(i)} - z^* \right) \right)$$
$$= \frac{p(i)\ell(i)}{\sqrt{t} + \sqrt{p(i)}\ell(i)} \left( \ell(i)\sqrt{p(i)} + \left( \ell(i) + \frac{\sqrt{t}}{\sqrt{p(i)}} \right) z^* - \sqrt{t} \right)$$
$$= \frac{p(i)\ell(i)}{\sqrt{t} + \sqrt{p(i)}\ell(i)} \ell(i)\sqrt{p(i)} = \sqrt{p(i)}\frac{\left( \sqrt{p(i)}\ell(i) \right)^2}{\sqrt{t} + \sqrt{p(i)}\ell(i)} = \sqrt{tp(i)}\zeta \left( \frac{\sqrt{p(i)}\ell(i)}{\sqrt{t}} \right). \tag{40}$$

In a similar way to that for showing (40), we can show

$$\ell(i)(p(i) - q(i)) - \sqrt{t}d^{(2)}(1 - q(i), 1 - p(i))$$
$$= -\ell(i)((1 - p(i)) - (1 - q(i))) - \sqrt{t}d^{(2)}(1 - q(i), 1 - p(i))$$
$$\leq \sqrt{t(1 - p(i))}\zeta\left(-\frac{\sqrt{1 - p(i)}\ell(i)}{\sqrt{t}}\right). \tag{41}$$

Combining (35), (36), (37), (40) and (41), we obtain (31). □

Note that $\xi(x)$ and $\zeta(x)$ defined in (32) satisfy $\xi(x) \leq x^2$ for $x \geq -1$ and $\zeta(x) \leq 2x^2$ for $x \geq -\frac{1}{2}$.

Using Lemma 9, we evaluate $\left\langle \hat{\ell}_t, q_t - q_{t+1} \right\rangle - D_t(q_{t+1}, q_t)$. As we define $p_t$ by (3) with $U = D$, we have $p_t(i) \geq \frac{\gamma_t}{|D|}$ for all $i \in D$. Hence, for any $i \in V_1 = \bigcup_{j \in D} N^{\text{out}}(j)$, the value of $P_t(i)$ defined by in (4) is bounded as

$$P_t(i) = \sum_{j \in N^{\text{in}}(i)} p_t(j) \geq \frac{\gamma_t}{|D|}, \tag{42}$$

which implies $\hat{\ell}_t \leq \frac{\ell_t(i)}{P_t(i)} \leq \frac{|D|}{\gamma_t}$. From this and the assumption of $\gamma_t \geq \frac{2|D|}{\beta_t}$, we have $\frac{\hat{\ell}_t(i)}{\beta_t} \leq \frac{|D|}{\beta_t\gamma_t} \leq \frac{1}{2}$ for all $i \in V_1$. As we have $\zeta(x) \leq x^2$ for $x \leq -\frac{1}{2}$, it holds for any $i \in V_1$ that

$$\mathbf{E}\left[\min\left\{q_t(i)\xi\left(\frac{\hat{\ell}_t(i)}{\beta_t}\right), (1 - q_t(i))\xi\left(-\frac{\hat{\ell}_t(i)}{\beta_t}\right)\right\}\right]$$

$$\leq \mathbf{E}\left[\min\left\{q_t(i), (1 - q_t(i))\right\}\left(\frac{\hat{\ell}_t(i)}{\beta_t}\right)^2\right]$$

$$= \mathbf{E}\left[\min\left\{q_t(i), (1 - q_t(i))\right\}\left(\frac{\ell_t(i)^2\mathbf{1}\left[i \in N^{\text{out}}(I_t)\right]}{P_t(i)^2\beta_t}\right)^2\right]$$

$$= \mathbf{E}\left[\min\left\{q_t(i), (1 - q_t(i))\right\}\frac{\ell_t(i)^2}{P_t(i)\beta_t^2}\right] \leq \mathbf{E}\left[\frac{2|D|}{\beta_t^2\gamma_t}q_t(i)(1 - q_t(i))\right],$$

where the last inequality follows from (42) and the inequality $\min\{x, 1 - x\} \leq 2x(1 - x)$ that holds for any $x \in [0, 1]$. We hence have

$$\mathbf{E}\left[\sum_{i \in V_1}\min\left\{q_t(i)\xi\left(\frac{\hat{\ell}_t(i)}{\beta_t}\right), (1 - q_t(i))\xi\left(-\frac{\hat{\ell}_t(i)}{\beta_t}\right)\right\}\right] \leq \mathbf{E}\left[\frac{2|D|}{\beta_t\gamma_t}\sum_{i \in V_1}q_t(i)(1 - q_t(i))\right]$$

$$= \mathbf{E}\left[\frac{2|D|b_t}{\beta_t^2\gamma_t}\right]. \tag{43}$$

For any $i \in V_2$, we have $i \in N^{\text{in}}(i)$, which implies $P_t(i) \geq p_t(i) \geq (1 - \gamma_t)q_t(i) \geq \frac{1}{2}q_t(i)$. We hence have

$$\mathbf{E}\left[\zeta\left(\frac{\sqrt{q_t(i)}\hat{\ell}_t(i)}{\sqrt{t}}\right)\right] \leq \mathbf{E}\left[\zeta\left(\frac{\sqrt{q_t(i)}\hat{\ell}_t(i)}{\sqrt{t}}\right)\right] \leq \mathbf{E}\left[\left(\frac{\sqrt{q_t(i)}\hat{\ell}_t(i)}{\sqrt{t}}\right)^2\right]$$

$$= \mathbf{E}\left[\frac{q_t(i)}{t}\frac{\ell_t(i)^2\mathbf{1}[i \in N^{\text{out}}(I_t)]}{P_t(i)^2}\right] \leq \mathbf{E}\left[\frac{q_t(i)}{tP_t(i)}\right] \leq \frac{2}{t}. \tag{44}$$

Further, if $q_t(i) \geq \frac{15}{16}$, we have $\frac{\sqrt{1 - q_t(i)}\hat{\ell}_t(i)}{\sqrt{t}} \leq \frac{1}{4P_t(i)\sqrt{t}} \leq \frac{1}{2q_t(i)\sqrt{t}} \leq \frac{8}{15}$. As $\zeta(x)$ satisfies $\zeta(x) \leq \frac{x^2}{1+x} \leq \frac{15}{7}x^2$ for any $x \geq -\frac{8}{15}$, we have

$$\zeta\left(-\frac{\sqrt{1 - q_t(i)}\hat{\ell}_t(i)}{\sqrt{t}}\right) \leq \frac{15}{7}\left(\frac{\sqrt{1 - q_t(i)}\hat{\ell}_t(i)}{\sqrt{t}}\right)^2 = \frac{15}{7}\frac{1 - q_t(i)}{t}\frac{\ell_t(i)^2\mathbf{1}[i \in N^{\text{out}}(I_t)]}{P_t(i)^2}$$

$$\leq \frac{60}{7}\frac{1 - q_t(i)}{t}\frac{\mathbf{1}[i \in N^{\text{out}}(I_t)]}{q_t(i)^2} \leq \frac{60}{7}\left(\frac{16}{15}\right)^2\frac{1 - q_t(i)}{t} \leq 10\frac{1 - q_t(i)}{t} \tag{45}$$

if $i \in V_2$ and $q_t(i) \geq \frac{15}{16}$. From (44) and (45), for $i \in V_2$, we have

$$\mathbf{E}\left[\min\left\{\sqrt{q_t(i)}\zeta\left(\frac{\sqrt{q_t(i)}\hat{\ell}_t(i)}{\sqrt{t}}\right), \sqrt{1-q_t(i)}\zeta\left(-\frac{\sqrt{1-q_t(i)}\hat{\ell}_t(i)}{\sqrt{t}}\right)\right\}\Big|q_t(i)\right]$$

$$\leq \begin{cases} 2\frac{\sqrt{q_t(i)}}{t} & \left(q_t(i) < \frac{15}{16}\right) \\ 10\frac{1-q_t(i)}{t} & \left(q_t(i) \geq \frac{15}{16}\right) \end{cases} = O\left(\frac{1}{t}\sqrt{q_t(i)(1-q_t(i))}\right). \tag{46}$$

We further have

$$\sum_{t=1}^{T}\left(\psi_t(q_{t+1}) - \psi_{t+1}(q_{t+1})\right) + \psi_{T+1}(\mu_{i^*}) - \psi_1(q_1)$$

$$= \sum_{i \in V_1}\left(\sum_{t=1}^{T}(\beta_t - \beta_{t+1})h(q_{t+1}(i))\right) + \sum_{i \in V_2}\left(\sum_{t=1}^{T}\left(\sqrt{t} - \sqrt{t+1}\right)g(q_{t+1}(i))\right)$$

$$\quad - 2\sqrt{T+1}\cdot|V_2| + \beta_1\sum_{i \in V_1}h(q_1(i)) + 2\sum_{i \in V_2}g(q_1(i))$$

$$= \sum_{t=1}^{T}(\beta_t - \beta_{t+1})a_{t+1} + 2\sum_{i \in V_2}\left(\sum_{t=0}^{T}\left(\sqrt{t+1} - \sqrt{t}\right)\left(\sqrt{q_{t+1}(i)} + \sqrt{1-q_{t+1}(i)} - 1\right)\right) + \beta_1 a_1$$

$$\leq \sum_{t=1}^{T}(\beta_t - \beta_{t+1})a_{t+1} + \beta_1 a_1 + 2\sum_{t=1}^{T+1}\frac{1}{\sqrt{t}}\sum_{i \in V_2}\sqrt{q_t(i)(1-q_t(i))}, \tag{47}$$

where $a_t$ and $b_t$ are defined by (16) and the last inequality follows from $\sqrt{t+1} - \sqrt{t} \leq \frac{1}{\sqrt{t+1}}$ and $\sqrt{x} + \sqrt{1-x} - 1 \leq \sqrt{x(1-x)}$. From Lemma 1 combined with (43), (46) and (47), we have

$$R_T = O\left(\sum_{t=1}^{T}\left(\gamma_t + \frac{|D|b_t}{\beta_t\gamma_t} + (\beta_t - \beta_{t+1})a_{t+1} + \frac{1}{\sqrt{t}}\sum_{i \in V_2}\sqrt{q_t(i)(1-q_t(i))}\right) + \beta_1 a_1\right). \tag{48}$$

### A.6   Proof of Proposition 2

*Proof.* We note that $b_t \leq 1$ and $b_t \leq a_t \leq 2\ln K$. We define $z_t = \frac{b_t a_{t+1}}{\gamma_t'}$ and $Z_t = \sum_{s=1}^{t}z_s$. Then, from the definition of $\gamma_t'$, we have

$$z_t = \frac{a_{t+1}b_t}{\gamma_t'} = 4\frac{a_{t+1}}{c_1}\left(c_1 + B_t^{1/3}\right) \geq a_{t+1} \geq b_{t+1} \tag{49}$$

where the second inequality follows from $b_t \leq a_t$. Further, we have

$$z_t = 4\frac{a_{t+1}}{c_1}\left(c_1 + B_t^{1/3}\right) \leq 4\left(c_1 + B_t^{1/3}\right) \leq 4c_1 + 4\left(b_1 + \sum_{s=1}^{t-1}z_s\right)^{1/3} \leq 8\left(c_1 + Z_{t-1}\right), \tag{50}$$

where the first inequality follows from $a_{t+1} \leq 2\ln K$ and $c_1 \geq 2\ln K$ and the last inequality follows from $c_1 \geq 2$ and $b_1 \leq 1$. From this, we have

$$\sum_{t=1}^{T}(\beta_{t+1} - \beta_t)a_{t+1} = c_2\sum_{t=1}^{T}\frac{z_t}{\sqrt{c_1 + Z_{t-1}}} = 4c_2\sum_{t=1}^{T}\frac{Z_t - Z_{t-1}}{3\sqrt{c_1 + Z_{t-1}} + \sqrt{c_1 + Z_{t-1}}}$$

$$\leq 4c_2\sum_{t=1}^{T}\frac{Z_t - Z_{t-1}}{\sqrt{c_1 + Z_t} + \sqrt{c_1 + Z_{t-1}}} = 4c_2\sum_{t=1}^{T}\left(\sqrt{c_1 + Z_t} - \sqrt{c_1 + Z_{t-1}}\right) \leq 4c_2\sqrt{Z_T}, \tag{51}$$

where the first equality follows from the definitions of $\beta_t$ and $z_t$, and the first inequality follows from (50).

We define $w_t = \frac{b_t}{\gamma_t'}$ and $W_t = \sum_{s=1}^{t} w_s$. From the definition of $\gamma_t'$, we have

$$w_t = \frac{b_t}{\gamma_t'} = 4\left(1 + \frac{1}{c_1} B_t^{1/3}\right) \geq 4. \tag{52}$$

Further, we have

$$w_1 \leq 8, \quad w_{t+1} = 4\left(1 + \frac{1}{c_1} B_{t+1}^{1/3}\right) \leq 4\left(1 + \frac{1}{c_1}(B_t + 1)^{1/3}\right) \leq 2w_t, \quad w_t \leq 4\left(1 + t^{1/3}\right). \tag{53}$$

Then $\beta_t$ can be bounded as

$$\beta_t = c_2 + c_2 \sum_{s=1}^{t-1} \frac{w_s}{\sqrt{c_1 + Z_{s-1}}} \geq \frac{c_2}{\sqrt{c_1 + Z_t}}\left(1 + \sum_{s=1}^{t-1} w_s\right)$$

$$= \frac{c_2}{\sqrt{c_1 + Z_t}}(1 + W_{t-1}) \geq \frac{c_2 t}{\sqrt{c_1 + Z_t}},$$

where the second inequality follows from (52). Hence, we have

$$\sum_{t=1}^{T} \frac{b_t}{\gamma_t \beta_t} \leq \sum_{t=1}^{T} \frac{b_t}{\gamma_t' \beta_t} \leq \sum_{t=1}^{T} \frac{\sqrt{c_1 + Z_t}}{c_2} \frac{w_t}{1 + W_{t-1}} \leq \frac{\sqrt{c_1 + Z_T}}{c_2} \sum_{t=1}^{T} \frac{w_t}{1 + W_{t-1}} \tag{54}$$

$$\leq O\left(\frac{\sqrt{c_1 + Z_T}}{c_2} \ln(1 + W_T)\right) \leq O\left(\frac{\sqrt{c_1 + Z_T}}{c_2} \ln T\right), \tag{55}$$

where the last inequality follows from (53) and the fourth inequality can be shown by taking the sum of the following inequality:

$$\ln(1 + W_t) - \ln(1 + W_{t-1}) = \ln\frac{1 + W_t}{1 + W_{t-1}} = \ln\left(1 + \frac{w_t}{1 + W_{t-1}}\right) \geq \frac{1}{4} \cdot \frac{w_t}{1 + W_{t-1}},$$

where the inequality follows from the facts that $\ln(1 + x) \geq \frac{1}{4}x$ holds for any $x \in [0, 8]$ and that (53) implies $\frac{w_t}{1+W_{t-1}} \leq 8$ for all $t$. We further have

$$\sum_{t=1}^{T} \frac{1}{\beta_t} \leq \sum_{t=1}^{T} \frac{\sqrt{c_1 + Z_t}}{c_2 t} \leq \frac{\sqrt{c_1 + Z_T}}{c_2} \sum_{t=1}^{T} \frac{1}{t} = O\left(\frac{\sqrt{c_1 + Z_T}}{c_2} \ln T\right). \tag{56}$$

In addition, we have

$$\sum_{t=1}^{T} \gamma_t' \leq \sum_{t=1}^{T} \frac{b_t}{c_1 + B_t^{1/3}} \leq \frac{3c_1}{2} \sum_{t=1}^{T} \left(B_t^{2/3} - B_{t-1}^{2/3}\right) \leq \frac{3c_1}{2} B_T^{2/3} \tag{57}$$

where the first inequality follows from $y^{2/3} - x^{2/3} \geq \frac{2}{3}(y - x)y^{-1/3}$, which holds for any $y \geq x > 0$. Combining (51), (55), (56) and (57), we obtain

$$\sum_{t=1}^{T} \left(\gamma_t + \frac{\delta b_t}{\gamma_t \beta_t} + (\beta_{t+1} - \beta_t)a_{t+1}\right) = \sum_{t=1}^{T} \left(\gamma_t' + \frac{2\delta}{\beta_t} + \frac{\delta b_t}{\gamma_t \beta_t} + (\beta_{t+1} - \beta_t)a_{t+1}\right)$$

$$= O\left(c_1 B_T^{2/3} + \left(\frac{\delta \ln T}{c_2} + c_2\right)\sqrt{c_1 + Z_T}\right)$$

$$= O\left(c_1 B_t^{2/3} + \left(\frac{\delta \ln T}{c_2} + c_2\right)\sqrt{c_1 + \sum_{t=1}^{T} \frac{a_{t+1}}{c_1}\left(c_1 + B_t^{1/3}\right)}\right)$$

$$= O\left(c_1 B_t^{2/3} + \frac{1}{\sqrt{c_1}}\left(\frac{\delta \ln T}{c_2} + c_2\right)\sqrt{c_1^2 + (\ln K + A_T)\left(c_1 + B_T^{1/3}\right)}\right),$$

where the third equality follows from (49) and the last equality follows from $a_{T+1} = O(\ln K)$. $\quad\square$

## A.7 Proof of Lemma 6

*Proof.* From the definition of $h(x)$, it holds for any $p \in \mathcal{P}(V)$ and $i^* \in [K]$ that

$$
-\sum_{i \in V_1} h(p(i)) \leq -\sum_{i \in V} h(p(i)) = \sum_{i \in V} \left( p(i) \ln \frac{1}{p(i)} + (1 - p(i)) \ln \frac{1}{1 - p(i)} \right)
$$

$$
= H(p) + \sum_{i \in V} (1 - p(i)) \ln \frac{1}{1 - p(i)} \leq (1 - p(i^*)) \ln \frac{eK}{1 - p(i^*)} + \sum_{i \in V} (1 - p(i)) \ln \frac{1}{1 - p(i)},
\tag{58}
$$

where the last inequality follows from (30). We further have

$$
\sum_{i \in V} (1 - p(i)) \ln \frac{1}{1 - p(i)} = (1 - p(i^*)) \ln \frac{1}{1 - p(i^*)} + \sum_{i \in V \setminus \{i^*\}} (1 - p(i)) \ln \left( 1 + \frac{p(i)}{1 - p(i)} \right)
$$

$$
\leq (1 - p(i^*)) \ln \frac{1}{1 - p(i^*)} + \sum_{i \in V \setminus \{i^*\}} (1 - p(i)) \left( \frac{p(i)}{1 - p(i)} \right) = (1 - p(i^*)) \left( \ln \frac{1}{1 - p(i^*)} + 1 \right).
\tag{59}
$$

Combining (58) and (59), we obtain

$$
-\sum_{i \in V_1} h(p(i)) \leq 2(1 - p(i^*)) \ln \frac{eK}{1 - p(i^*)}.
$$

From this, we have

$$
A_T = -\sum_{t=1}^{T} \sum_{i \in V_1} h(q_t(i)) \leq 2 \sum_{t=1}^{T} (1 - q_t(i^*)) \ln \frac{eK}{1 - q_t(i^*)} \leq 2Q(i^*) \ln \frac{eKT}{Q(i^*)},
$$

where the last inequality follows from the similar argument to Lemma 4. We also have

$$
B_T \leq \sum_{t=1}^{T} \sum_{i \in V} q_t(i)(1 - q_t(i)) = \sum_{t=1}^{T} \left( q(i^*)(1 - q_t(i^*)) + \sum_{i \in V \setminus \{i^*\}} q_t(i)(1 - q_t(i^*)) \right)
$$

$$
\leq \sum_{t=1}^{T} \left( (1 - q_t(i^*)) + \sum_{i \in V \setminus \{i^*\}} q_t(i) \right) = 2 \sum_{t=1}^{T} (1 - q_t(i^*)) = 2Q(i^*).
$$

for any $i^* \in [K]$. This completes that proof of Lemma 6. $\qquad \square$

## A.8 Proof of Lemma 7

*Proof.* We have

$$
\sum_{t=1}^{T} \frac{1}{\sqrt{t}} \sum_{i \in V_2} \sqrt{q_t(i)(1 - q_t(i))} \leq \sum_{t=1}^{T} \frac{1}{\sqrt{t}} \sqrt{|V_2| \sum_{i \in V_2} q_t(i)(1 - q_t(i))}
$$

$$
\leq \sqrt{\left( \sum_{t=1}^{T} \frac{1}{t} \right) \left( |V_2| \sum_{t=1}^{T} \sum_{i \in V_2} q_t(i)(1 - q_t(i)) \right)} \leq \sqrt{|V_2|(\ln T + 1) \sum_{t=1}^{T} \sum_{i \in V_2} q_t(i)(1 - q_t(i))},
\tag{60}
$$

where inequalities follow from the Cauchy-Schwarz inequality. We further have

$$
\sum_{t=1}^{T} \sum_{i \in V_2} q_t(i)(1 - q_t(i)) \leq \sum_{t=1}^{T} (1 - q_t(i^*)) + \sum_{t=1}^{T} \sum_{i \in V_2 \setminus \{i^*\}} q_t(i) \leq 2 \sum_{t=1}^{T} (1 - q_t(i^*)) = 2Q(i^*)
$$

for any $i^* \in [K]$. Combining this with (60), we obtain $R_T^{(2)} = O\left( \sqrt{|V_2| \ln T \cdot Q} \right)$. $\qquad \square$

# B    Comparison with the result by Rouyer et al. [2022]

While Rouyer et al. [2022] consider the same research question as this paper, their approach is different from ours in the following points. Their algorithm follows the approach by Seldin and Slivkins [2014] and Seldin and Lugosi [2017], in which the suboptimality gaps $\Delta_i$ are explicitly estimated. In contrast, our algorithms do not use explicit estimation for suboptimality gap, and instead employ the self-bounding technique to lead to stochastic regret bounds, similarly to the algorithms by Zimmert and Seldin [2021], Wei and Luo [2018]. Due to these differences in algorithm design and regret analysis, it seems difficult to integrate these algorithms or provide a unified analysis.

The differences in results can be summarized as follows:

- Advantage of our results:
    - Covered classes of feedback graphs: We provide algorithms for both strongly observable graphs and weakly observable graphs. On the other hand, the algorithms by Rouyer et al. [2022] only deal with graphs with self-loops, which is a special case of strongly observable graphs.
    - Our algorithms can also handle stochastic environments with adversarial corruptions.
    - Our regret bounds for strongly observable graph depend on the independence number $\alpha$ while the algorithms by Rouyer et al. [2022] depend on strong independent number $\tilde{\alpha}$, which is the independence number of the subgraph consisting of bidirectional edges. In general $\alpha \leq \tilde{\alpha}$, and for symmetric graphs $\alpha = \alpha'$. We also note that, in some cases, there is a significant discrepancy between $\alpha$ and $\tilde{\alpha}$. For example, a directed graph $G = (V, E)$ defined by $V = [K]$, $E = \{(i, j) \in V \times V \mid i \leq j\}$ has $\alpha = 1$ and $\tilde{\alpha} = K$.
- Advantage of results by Rouyer et al. [2022]:
    - Their algorithm has a regret bound expressed with individual suboptimality gaps $\Delta_i$ for stochastic environments, while the regret bounds in this paper depend only on $\Delta_{\min} = \min_{i \in [K] \setminus \{i^*\}} \Delta_i$. Consequently, if many actions $i$ have large suboptimality gaps $\Delta_i \gg \Delta_{\min}$, their algorithms will perform better.
    - Their regret bound has an improved dependency on $\ln T$. More precisely, their stochastic regret bounds for problems with strongly observable graphs scale with $O((\ln T)^2)$, which is better than our regret bounds of $O((\ln T)^3)$.
    - Their paper includes extension to time varying feedback graphs though our algorithms seem to be extendable in a similar way.

# C    An alternative algorithm for the weakly observable case

In the weakly observable case, as shown in Theorem 2, our regret bound for stochastic environments include an $O(\frac{K' \ln T}{\Delta_{\min}})$-term, where $K' = |V_2|$ is the number vertices that are not dominated by the weakly dominating set $D$. When $T$ is sufficiently larger than other problem parameters, this term is negligibly small compared to the other term $\frac{\delta (\ln T)^2}{\Delta_{\min}^2}$. However, if $K'$ is larger than $\frac{\delta \ln T}{\Delta_{\min}}$, this $O(\frac{K' \ln T}{\Delta_{\min}})$-term can be dominant. In such a case, the regret upper bound may be improved by modifying the algorithm. Roughly speaking, by combining the approach to strongly observable case, the $O(\frac{K' \ln T}{\Delta_{\min}})$-term can be replaced with an $O(\frac{\alpha^{(2)} (\ln T)^3}{\Delta_{\min}})$-term, where $\alpha^{(2)}$ is the independent number of the subgraph induced by $V_2$, i.e.,

$$G_2 = (V_2, E \cap (V_2 \times V_2)), \quad \alpha^{(2)} = \alpha(G_2). \tag{61}$$

We here note that $G_2$ is a strongly observable graph with self-loops as $D$ is a weakly dominating set (Definition 2) and $V_2 = V \setminus \bigcup_{i \in D} N^{\text{out}}(i)$. If $\alpha^{(2)} (\ln T)^2 \leq K'$, the modified version provides a better regret bound. The details of the modification are given below.

Consider the following regularizer function:

$$\psi_t(p) = \beta_t^{(1)} \sum_{i \in V_1} h(p(i)) + \beta_t^{(2)} \sum_{i \in V_2} h(p(i)), \quad \text{where} \quad h(x) = x \ln x + (1 - x) \ln(1 - x).$$

We define $\beta_t^{(1)}$ and $\gamma_t^{(1)}$ in the same way as (15) in Section 6 with repracement of $c_1 := c_1^{(1)}$ and $c_2 := c_2^{(1)}$. Similarly, we define $\beta_t^{(2)}$ and $\gamma_t^{(2)}$ in a similar way as (9) in Section 5 with $c_1 := c_1^{(2)}$ and $a_s := \sum_{i \in V_2} h(q_s(i))$. Parameters $c_1^{(1)}$, $c_2^{(1)}$ and $c_1^{(2)}$ are specified later. Using this regularizer function, we compute $q_t$ using FTRL given by (3). Then, we compute $p_t$ by

$$p_t = (1 - \gamma_t^{(1)} - \gamma_t^{(2)})q_t + \gamma_t^{(1)}\mu_D + \gamma_t^{(2)}\mu_{V_2}. \tag{62}$$

We then have the following regret bound:

$$R_T \le \hat{c}^{(1)} \cdot \max\left\{ \bar{Q}^{2/3}, \left(c_1^{(1)}\right)^2 \right\} + \hat{c}^{(2)} \cdot \max\left\{ \bar{Q}^{1/2}, 1 \right\} \quad \text{where}$$

$$\hat{c}^{(1)} = O\left( c_1^{(1)} + \frac{1}{\sqrt{c_1^{(1)}}}\left( \frac{|D|\ln T}{c_2^{(1)}} + c_2^{(1)} \right)\sqrt{\ln(KT)} \right),$$

$$\hat{c}^{(2)} = O\left( \left( \frac{\alpha^{(2)}\ln T \cdot \ln(c_1^{(2)}KT)}{c_1^{(2)}\sqrt{\ln K}} + c_1^{(2)}\sqrt{\ln K} \right)\sqrt{\ln(KT)} \right). \tag{63}$$

Consequently, in adversarial regimes with self-bounding constraints, we have

$$R_T = O\left( \frac{(\hat{c}^{(1)})^3}{\Delta_{\min}^2} + \left( \frac{C^2(\hat{c}^{(1)})^3}{\Delta_{\min}^2} \right)^{1/3} + O\left( \frac{(\hat{c}^{(2)})^2}{\Delta_{\min}} + \sqrt{\frac{C(\hat{c}^2)^2}{\Delta_{\min}}} \right) \right). \tag{64}$$

Similarly to the analysis in Section 6, we obtain $\hat{c}^{(1)} = O\left( (|D|\ln T \cdot \ln(KT))^{1/3} \right)$ by setting $c_1^{(1)} = \Theta\left( (|D|\ln T \cdot \ln(KT))^{1/3} \right)$ and $c_2^{(1)} = \Theta\left( \sqrt{|D|\ln T} \right)$. Further, by setting $c_1^{(2)} = \Theta\left( \sqrt{\frac{\alpha^{(2)}\ln T \cdot \ln(KT)}{\ln K}} \right)$, we obtain $\hat{c}^{(2)} = O\left( \sqrt{\alpha^{(2)}\ln T \cdot (\ln(KT))^2} \right)$.

Consequently, the modified algorithm achieves

$$R_T = |D|^{1/3}(T\ln T)^{2/3} + \sqrt{\alpha^{(2)}T(\ln T)^3} \tag{65}$$

for adversarial environments and

$$R_T = \frac{|D|(\ln T)^2}{\Delta_{\min}^2} + \left( \frac{C^2|D|(\ln T)^2}{\Delta_{\min}^2} \right)^{1/3} + \frac{\alpha^{(2)}(\ln T)^3}{\Delta_{\min}} + \left( \frac{C\alpha^{(2)}(\ln T)^3}{\Delta_{\min}} \right)^{1/2} \tag{66}$$

for stochastic environments with adversarial corruptions (more generally, in adversarial regimes with self-bounding constraints).

## D  Note on the definition of weak domination

Previous studies, e.g., Alon et al. [2015], have adopted a slightly different definition of *weak domination* rather than one in this paper:

**Definition 4** (alternative difitnition of weak domination, [Alon et al., 2015])**.** For any directed graph $G = (V, E)$ with a set of weakly observable vertices $W \subseteq V$, a *weakly observable set* $D' \subseteq V$ is a set of vertices that dominates $W$, i.e., that satisfies $W \subseteq \bigcup_{i \in D'} N^{\text{out}}(i)$. The *weak domination number* $\delta'(G)$ of G is the size of its smallest weakly dominating set.

We can see that our definition of weakly dominating set in Definition 2 and that in Definition 4 coincide, with some very limited exceptions. Consequently, we will see that $\delta(G)$ and $\delta'(G)$ in Definitions 2 and 4 satisfy $\delta(G) \le \delta'(G) \le \delta(G) + 1$. Further, if $\delta(G) \ge 2$ then $\delta(G) = \delta'(G)$. These facts can be confirmed as follows.

From the definition observability (Definition 1), the vertices of a weakly observable graph are classified into the following three type:

**strongly observable vertices, type 1** $V_{\text{SO1}} = \{i \in V \mid i \in N^{\text{in}}(i)\}$: vertices with self-loop (strongly observable vertices, type 1).

**strongly observable vertices, type 2** $V_{\text{SO2}} = \{i \in V \mid N^{\text{in}}(i) = V \setminus \{i\}\}$: vertices without self-loop, with edges from all other vertices. (strongly observable vertices, type 2).

**weakly observable vertices** $V_{\text{WO}} = V \setminus (V_{\text{SO1}} \cup V_{\text{SO2}})$: weakly observable vertices.

Weakly dominating set $D$ in Definition 2 dominates all vertices except $V_{\text{SO1}}$, i.e., all vertices in $V_{\text{SO2}} \cup V_{\text{WO}}$. Weakly dominating set $D'$ in Definition 4 dominates $V_{\text{WO}}$. It is clear that $D'$ dominates $V_{\text{SO1}}$, which means that $D'$ is a weakly dominating set in the sense of Definition 2 as well. On the other hand, if the size of $D$ is greater than or equal to 2, then it also dominates all vertices $V_{\text{SO2}}$. This implies that $D$ is a weakly dominating set in the sense of Definition 4 as well. Therefore, for vertex sets of size at least 2, the concept of weak domination is the same in Definition 2 as in Definition 4. Consequently, we have $\delta(G) = \delta'(G)$ if $\delta(G) \geq 2$.

The only exception is the case in which $|D| = 1$ and $D \subseteq V_{\text{SO2}}$. In this case, however, by adding an arbitrary vertex to $D$, we can make it dominate $V_{\text{SO2}}$ as well. In other words, for any $i \in V \setminus D$, $D \cup \{i\}$ dominates $V_{\text{SO2}}$, and hence, is a weakly dominating set in the sense of Definition 4 as well. Hence, even if $\delta(G) = 1$, we have $1 \leq \delta'(G) \leq 2$.