# OpenReview forum: "Nearly Optimal Best-of-Both-Worlds Algorithms for Online Learning with Feedback Graphs"
_NeurIPS.cc/2022/Conference — NeurIPS 2022 Accept_

### Official Review · Reviewer_1CHF · 2022-07-07

**Rating:** 7
**Confidence:** 4
**Soundness:** 4 excellent
**Presentation:** 3 good
**Contribution:** 3 good

**Summary:**

This paper considers the design of best-of-both-worlds algorithms for the problem of online learning with feedback graphs. Online learning is a sequential setting which proceeds in rounds $t = 1, \ldots, T$, where in each round the learner issues a prediction, suffers a loss that corresponds to this prediction, and then obtains some feedback on the prediction. The goal is to control the regret, which is the difference between the cumulative loss of the learner and the cumulative loss of the best fixed prediction in hindsight. This version of online learning considers possibly partial feedback scenarios modelled by directed graphs where nodes are the available actions. In this setting, the feedback received by the learner after choosing an action is given by the loss of actions in the out-neighbourhood of the currently selected action. Best-of-both-worlds algorithms try to achieve optimal regret in both adversarial and stochastic scenarios simultaneously, where losses may or may not follow an underlying distribution. The authors extend previous work to more general feedback structures. They manage to design best-of-both-worlds algorithms for any strongly observable or weakly observable feedback graph, respectively, and prove near-optimal regret bounds for them. In particular, for strongly observable graphs with independence number $\alpha$, the bounds are of order $\sqrt{\alpha T}$ for adversarial losses and $\alpha \ln(T)^3/\Delta$ for stochastic losses, where $\Delta$ is the difference between the expected loss of the best arm and the second best arm. For weakly observable feedback graphs with domination number $\delta$, the bounds are of order $\delta^{1/3}T^{2/3}$ for adversarial losses and $\delta \ln(T)^2/\Delta^2$ for stochastic losses.

**Questions:**



**Limitations:**

The authors have adequately addressed the limitations and potential negative societal impact of their work.

**Strengths And Weaknesses:**

Overall, the paper is well written and the results are new. However, the authors should take some care in addressing the typo’s in the paper, of which some are listed below. The authors provide sufficient motivations for studying the problem at hand. Not only do the results improve upon previous work in terms of regret, they are also more general  as the feedback graphs in this paper are arbitrary feedback graphs rather than restricted feedback graphs considered in previous work. For stochastic losses, compared to the previously best known result the improvement in the strongly observable case is a factor ln(T)^2 and an improvement in the leading factor, which changes from the clique covering number to the independence number. For the weakly observable case there were no previous best of both world results. The near-optimality of the regret bound for the proposed algorithms required an improved analysis on the performance of follow-the-regularized-leader (FTRL) with the Shannon entropy regularizer, while using a novel definition for the update rule of the learning rate at the same time. The above results are indeed not immediate to achieve. As the authors themselves state, this might be useful for different applications of FTRL with a Shannon entropy regularizer too.

Relevance:
Best-of-both-worlds algorithms have been of recent interest in online learning settings. Therefore, the submitted work is sufficiently relevant since it provides significant advances addressing the online learning problem with feedback graphs.

Clarity:
The results are presented in a clear manner and are well contextualised in the literature.

Weaknesses:
In both the weakly and strongly observable case the upper bounds for stochastic losses may be a factor $\ln(T)^2$ away from being optimal. However, the upper bounds improve greatly on previous work, and the lower bound in the weakly observable case probably can be improved.

---

> ### Author Response · Authors · 2022-08-02
> **Response to Reviewer 1CHF**
>
> Thank you for taking your valuable time to review this paper.
> We will revise the paper as follows.
>
> > In both the weakly and strongly observable case the upper bounds for stochastic losses may be a factor $\ln (T)^2$ away from being optimal. However, the upper bounds improve greatly on previous work, and the lower bound in the weakly observable case probably can be improved.
>
> Improving the dependence of $\ln T$ in the regret bound is an important future work.
> Reviewer 7bnY pointed out that independent contemporaneous studies have shown better regret upper bounds for the dependence of $\ln T$.
> In the revised version, these facts and the description of future work will be added.

---

### Official Review · Reviewer_3PsX · 2022-07-10

**Rating:** 7
**Confidence:** 4
**Soundness:** 4 excellent
**Presentation:** 3 good
**Contribution:** 3 good

**Summary:**

The authors consider the online learning with directed feedback graphs problem and present algorithms based on the follow-the-regularized-leader approach, which achieve best-of-both worlds regret guarantees in both the fully observable and the weakly observable settings.

The regret bounds presented in the paper are optimal up to polylog factors i advnersarial settings, and are polylogarithmic in $T$ in stochastic settings. The authors also prove regret bounds for the more general adversarial regime with a self-bounding constraint.

In the fully observable case, the authors employ FTRL with a negative entropy regularization and an adaptive learning rate and exploration scheme to obtain regret bounds that scale with the feedback graph's independence number $\alpha$. In the weakly observable setting, the authors use a regularization which depends on a weakly dominating set, while employing a negative entropy regularization on vertices without self-loops, and Tsallis entropy on the vertices with self loops. In turn, the regret bounds obtained in the weakly observable case scales with the feedback graph's weak domination number $\delta$.

**Questions:**

* The definition of a weakly dominating set appearing in this paper seems to differ from the definition in previous works, such as Alon et al. [2015]. In their paper, a weakly dominating set is one which dominates the weakly observable vertices. i.e. those without a self loop which also don't have incoming edges from every other vertex. However, the authors define a weakly dominating set by one that dominates all the vertices with no self loops. This makes the weak domination number - according to the authors' definition - larger in the worst case than in the definition from previous works, and in turns makes the regret bounds the authors obtain for weakly observable graphs sub-optimal in another sense. I suggest the authors either match their definition of a weakly dominating set with the definitions from previous works and modify their analysis accordingly, or alternatively, discuss the discrepancy between the two definitions and the relation between the regret bounds obtained with each one.
* It is a bit unclear to me why the extra $\frac{K'}{\Delta}$ is necessary in the stochastic bound in the weakly observable setting. Since the set $V_2$ contains vertices with self-loops, i.e. fully observable vertices, instead of using Tsallis entropy on those vertices which seems to be what contributes this additive factor, why not just use the approach from the fully observable case? I.e, use negative entropy regularizer with the step size scheme described in Section 5. I'm not sure where the analysis breaks given this approach, as it seems that by Lemma 5 the regret breaks into a term which only depends on $V_1$ and a term which depends only on $V_2$ so both terms can presumably be handled separately. I would appreciate it if the authors could explain their choice of Tsallis on $V_2$ and why the approach I suggested wouldn't work.

**Limitations:**

The authors properly address most aspects regarding the limitations of their results, specifically the suboptimality of their regret bounds in stochastic regimes.
See the main review for remarks on other limitations which I believe were not properly addressed, such as the discrepancy between their definition of a weakly dominating set and previous definitions, and the required knowledge of the independence number in the fully observable setting.

**Strengths And Weaknesses:**

Strengths:
* The results presented in the paper improve upon currently known best-of-both-worlds regret bounds in the online learning with feedback graphs framework, and they are relevant in a very general setting of directed feedback graphs, as well as weakly observable graphs.
* In the fully observable setting, the authors show that FTRL with negative entropy regularization suffices in order to obtain nearly optimal regret bounds, by implementing a novel adaptive step size scheme. This shows that highly complex regularizers are not essential in order to obtain such regret bounds for feedback graphs, and consequently the technical analysis in the paper is very clean and concise.
* The authors present the first best-of-both-worlds regret bounds which hold for weakly observable graphs. while in the stochastic setting their bounds are sub-optimal, this result is highly non-trivial and is achieved with a simple algorithm which is an instantiation of FTRL with a regularizer that combines negative entropy and Tsallis entropy.

Weaknesses:
* The stochastic bounds presented in the paper are suboptimal in the following sense: For strongly observable graphs, the stochastic bound scales with $O(\frac{\alpha}{\Delta})$ which in some cases could be worse than the optimal bound which scales with $O(\sum_{i \in A} \frac{1}{\Delta_i})$ where the sum is over a maximum independent set, and $\Delta_i$ is the suboptimality gap of arm $i$. For weakly observable graphs, the stochastic bound includes an additive term which scales like $\frac{K'}{\Delta}$ where $K'$ is the number of arms with self-loops. In the worst case, this could scale linearly with $K$ which is known to be suboptimal.
* In the strongly observable setting, the algorithm requires the independence number of the feedback graph in order to appropriately tune the step sizes. Since computing the independence number of a graph is computationally hard, this presents a certain limitation of the algorithm. While this limitation also appears in previous works, I think the authors should mention the algorithm's dependence on $\alpha$.
* A few typos: In the abstract "follow-the-perturbed-leader" is mentioned instead of "follow-the-regularized-leader", in the statement of Lemma 1 the first appearance of $\psi_t(q_{t+1})$ should be replaced by $\psi_{t+1}(q_{t+1})$.

---

> ### Author Response · Authors · 2022-08-02
> **Response to Reviewer 3PsX**
>
> Thanks for your careful reading and many helpful suggestions.
> We will revise the paper based on your comments as follows.
>
> > The stochastic bounds presented in the paper are suboptimal in the following sense: For strongly observable graphs, the stochastic bound scales with $O(\frac{\alpha}{\Delta})$ which in some cases could be worse than the optimal bound which scales with O(\sum_{i \in A} \frac{1}{\Delta_i}) where the sum is over a maximum independent set, and $\Delta_i$ is the suboptimality gap of arm $i$. ...
>
> As pointed out, our $\Delta_{\min}$-dependent bounds are worse in some cases than the individual $\{ \Delta_i \}$-dependent bounds, and are suboptimal in that sense.
> The revised version will mention this point.
> The question regarding the possibility of $\{ \Delta_i \}$-dependent regret upper bounds with our approach is discussed in our response to Reviewer eXST.
>
> > In the strongly observable setting, the algorithm requires the independence number of the feedback graph in order to appropriately tune the step sizes. ..
>
> Indeed, computing the independence number is NP-hard.
> However, if we can obtain an approximate value, such as $\alpha'$ satisfying $\alpha \le \alpha'\le (1+\epsilon)\alpha$, by setting the learning rate accordingly, we have regret bounds with an additional $\sqrt{1 + \epsilon}$ factor for adversarial environments and with $(1 + \epsilon)$ factor for stochastic environments, respectively.
> As discussed in the last paragraph of Section 4 of [Alon et al., 2017], intricate graphs with hard-to-approximate independence numbers seem to be rare in real-world applications, and therefore a reasonable approximation can be computed using standard heuristics in many cases.
> A discussion of this will be added in the revised version.
>
> > A few typos: In the abstract "follow-the-perturbed-leader" is mentioned instead of "follow-the-regularized-leader", in the statement of Lemma 1 the first appearance of should be replaced by .
>
> Thanks for pointing out the typos.
> We will correct it in the revised version.
>
> > The definition of a weakly dominating set appearing in this paper seems to differ from the definition in previous works, such as Alon et al. [2015]. ...
>
> Thank you for pointing out the discrepancy of the definitions.
> Our definition of weakly dominating set and that of Alon et al. [2015] coincide, with some limited exceptions.
> Further, the difference in the weak domination numbers is at most 1, so it only affects the regret bounds by a constant factor.
>
>
> The vertices of a weakly observable graph are classified into the following three types:
> - SO: strongly observable vertices
>     - SO1: vertices with self-loop
>     - SO2: vertices without solf-loop, with edges from all other vertices
> - WO: weakly observable vertices
>
> Alon et al. [2015] defines a weakly dominating set as a vertex set  that dominates all of WO [Definition 1].
> Our paper defines a weakly dominating set as a vertex set  that dominates all vertices except SO1 (i.e., SO2 and WO) [Definition 2].
> It is clear that if a set S satisfies Definition 2, then it satisfies Definition 1.
> If S satisfies Definition 1 and the size of S is greater than or equal to 2, then it also satisfies Definition 2 because it dominates all SO2.
> The exception is the case where S satisfies definition 1 but is of size 1 and it is contained in SO2, which does not satisfy definition 2.
> In this case, however, by adding an arbitrary vertex to S, definition 2 is satisfied.
> Thus, the difference in the weak domination number defined in each paper is at most 1.
> In the revised version, we add a note on this difference.
>
> > It is a bit unclear to me why the extra $\frac{K'}{\Delta}$ is necessary in the stochastic bound in the weakly observable setting. ...
>
>
> We deeply appreciate your contribution of ideas for improving the algorithm.
> We had not thought of your approach.
> At this time, we have not found any points where the analysis breaks down in the suggested approach, and the results seem to be as expected by the reviewer.
> Once we have written down the proof and confirmed that there are no problems, we will add the suggested approach and the results obtained from it to the revised version.
>
>
> Reference:
>
> - N. Alon, N. Cesa-Bianchi, O. Dekel, and T. Koren. Online learning with feedback graphs: Beyond bandits. Journal of Machine Learning Research, 40(2015), 2015.
> - N. Alon, N. Cesa-Bianchi, C. Gentile, S. Mannor, Y. Mansour, and O. Shamir. Nonstochastic multi332 armed bandits with graph-structured feedback. SIAM Journal on Computing, 46(6):1785–1826, 2017.

---

### Official Review · Reviewer_7bnY · 2022-07-11

**Rating:** 7
**Confidence:** 5
**Soundness:** 4 excellent
**Presentation:** 3 good
**Contribution:** 3 good

**Summary:**

This paper studies the best-of-both-world problem for bandits with feedback graphs.
It derives an algorithm which is up to log factors optimal in the adversarial regime and obtains reasonable regret guarantees in the stochastic regime.
The stochastic bound is sub-optimal with regard to log factors and instance dependent complexity, but it is still outperforming the best-known bound for best-of-both-worlds algorithms.
Additionally, it proposes best-of-both worlds algorithms in the weakly observable aka T^2/3 regime.

**Questions:**

Are you aware of the extremely similar paper:
"A Near-Optimal Best-of-Both-Worlds Algorithm for Online Learning with Feedback Graphs" ?
Given that you seem to have obtained almost identical results with very similar techniques in concurrent work, it might make sense to combine the two papers?


**Limitations:**

Theoretical work, no societal impact.

**Strengths And Weaknesses:**

The math is sound and the presentation is clear.
There are technical novelties over the vanilla self-bounding regret analysis due to the fact that Shannon entropy is required for optimal regret in the adversarial regime.
An important limitation of this work is the existence of a very similar work (even up to the title): "A Near-Optimal Best-of-Both-Worlds Algorithm for Online Learning with Feedback Graphs".
The other work explicitly estimates the gaps in the stochastic regime and obtains a better stochastic bound. This work on the other side is purely using the self-bounding technique which simplifies the algorithm and allows for a straightforward bound for the corrupted stochastic setting. The analysis for weakly observable graphs is also unique to this paper here.
Overall, both paper build on the same literature and use very similar algorithmic techniques to obtain almost identical results.

---

> ### Author Response · Authors · 2022-08-02
> **Response to Reviewer 7bnY**
>
> Thank you for your time and for sharing information on a deeply relevant paper.
> We have described below the relationship between the papers and our views on them.
> We hope that what follows addresses your concerns.
>
> > An important limitation of this work is the existence of a very similar work (even up to the title): "A Near-Optimal Best-of-Both-Worlds Algorithm for Online Learning with Feedback Graphs". The other work explicitly estimates the gaps in the stochastic regime and obtains a better stochastic bound. This work on the other side is purely using the self-bounding technique which simplifies the algorithm and allows for a straightforward bound for the corrupted stochastic setting. The analysis for weakly observable graphs is also unique to this paper here. Overall, both paper build on the same literature and use very similar algorithmic techniques to obtain almost identical results.
>
> > Are you aware of the extremely similar paper: "A Near-Optimal Best-of-Both-Worlds Algorithm for Online Learning with Feedback Graphs" ? Given that you seem to have obtained almost identical results with very similar techniques in concurrent work, it might make sense to combine the two papers?
>
> We became aware of this paper after we submitted our paper.
> We agree with the reviewers that the results are very similar to our paper.
> On the other hand, as pointed out, their and our papers differ in that their algorithm explicitly estimates the suboptimality gap (an extension of Exp3++ [Seldin and Slivkins, 2014; Seldin and Lugosi, 2017]) while our algorithm does not thanks to the self-bounding technique (similar to Tsallis-INF [Zimmert and Seldin 2021] and BROAD [Wei and Luo 2018]).
> These two types of algorithms have very different proofs of the regret upper bound in the stochastic setting, which seems to make it difficult to integrate them or provide a unified analysis.
>
>
>
> Comparing the results, each of ours and theirs has the following advantages:
> - Advantages of our results:
> 	- Covered classes of feedback graphs: We provide algorithms for both strongly observable graphs and weakly observable graphs. On the other hand, they only deal with graphs with self-loops, which is a special case of strongly observable graphs.
> 	- Our algorithms can also handle stochastic environments with adversarial corruption.
> 	- Our regret bounds for strongly observable graph depends on the independence number $\alpha$, while theirs depend on *strong* independent number $\tilde{\alpha}$, which is the independence number of the subgraph consisting of bidirectional edges. In general $\alpha \le \tilde{\alpha}$, and for symmetric graphs $\alpha = \tilde{\alpha}$. We also note that, in some cases, there is a significant discrepancy between $\alpha$ and $\tilde{\alpha}$.
> 	For example,
> 	a directed graph $G = (V, E)$ defined by $V = [K]$, $E = \\{ (i, j) \in V \times V \mid i \le j \\}$ has $\alpha = 1$ and $\tilde{\alpha} = K$.
> - Advantages of their results
> 	- Their algorithm has a regret bound expressed with individual suboptimality gaps $\Delta_i$ for stochastic environments.
> 	- Their regret bound has an improved dependency with respect to $\log T$.
> 	- Their paper includes extension to time varying feedback graphs (our algorithm seems to be extendable in a similar way).
>
> Since each paper has its own strengths and weaknesses as described above, it is difficult to say in general which is superior to the other.
> It would be meaningful if both methods could be combined to achieve both strengths.
> In our side, we are ready to consider the possibility of combining the papers.
> After hearing the wishes of the other party, we would like to positively consider the possibility if they are willing to do so.
>
> Reference
>
> - Y. Seldin and G. Lugosi. An improved parametrization and analysis of the EXP3++ algorithm for stochastic and adversarial bandits. In Conference on Learning Theory, 2017.
> - Y. Seldin and A. Slivkins. One practical algorithm for both stochastic and adversarial bandits. In Proceedings of the International Conference on Machine Learning (ICML), 2014.
> - C.-Y. Wei and H. Luo. More adaptive algorithms for adversarial bandits. In Conference on Learning Theory, pages 1263–1291, 2018.
> - J. Zimmert and Y. Seldin. Tsallis-INF: An optimal algorithm for stochastic and adversarial bandits. Journal of Machine Learning Research, 22(28):1–49, 2021.

---

### Official Review · Reviewer_eXST · 2022-07-12

**Rating:** 8
**Confidence:** 4
**Soundness:** 3 good
**Presentation:** 4 excellent
**Contribution:** 4 excellent

**Summary:**

The authors consider the Best-of-Both-Worlds setting in Online Learning with Feedback Graph: the rewards of arms can be stochastic or adversarial and the feedback of them is described by a directed graph of arms. The authors show for strongly observable graphs, FTRL with the negative entropy regularizer and adaptive learning rates has optimal regret bounds up to logarithmic factors in both stochastic and adversarial settings. For weakly observable graphs, a more sophisticated regularizer give similar results.

**Questions:**

Can authors talk about whether it is possible to get bounds scaling with $\sum_{i\in S}\frac{1}{\Delta_i}$ for strongly observable graphs under this approach?

**Strengths And Weaknesses:**

- This is a great paper. The problem is important and challenging, but the authors give pretty simple and clear solutions. For strongly observable graphs, it is not surprising that entropy can be used for this setting, but how to properly tune the learning rates seems highly nontrivial. I roughly checked the proof and didn't find any issues. Overall the analysis is quite concise. For weakly observable graphs, the choice of the regularizer is more mysterious to me. Maybe the authors can comment more about this, for example, the intuition to add $(1-x)\ln(1-x)$ and $\sqrt{1-x}$ terms.
- Besides suboptimal orders of logarithmic terms, one weakness is that the gap-dependent constants are still not optimal. For example, consider standard bandit feedback. The bound in this case is $\frac{\alpha(\ln T)^3}{\Delta_{\min}}$, which is not ideal once K is very large and there is only one small gap. I think it would be great to comment this somewhere or show some bounds of optimal stochastic algorithms in the table.

---

> ### Author Response · Authors · 2022-08-02
> **Response to Reviewer eXST**
>
> Thank you for your time and helpful suggestions.
> We hope the following answers address your questions and concerns.
>
> > For weakly observable graphs, the choice of the regularizer is more mysterious to me. Maybe the authors can comment more about this, for example, the intuition to add $(1-x) \ln (1-x)$ and $\sqrt{1-x}$ terms.
>
> The regularization of $(1-x) \ln (1-x)$ and $\sqrt{1-x}$ is introduced to apply the self-bounding technique.
> Without such regularization, the resulting regret upper bound would include the arm-selection probability $q_{ti^*}$ of the optimal arm $i^*$, which is inconvenient for the self-bounding technique.
> To apply the self-bounding technique, we need a regret upper bound expressed in terms of the arm-selection probability $q_{ti}$ of the non-optimal arm $i \in [K] \setminus \{ i^* \}$.
> By adding the regularization of $(1-x) \ln (1-x)$ and $\sqrt{1-x}$, $q_{ti^*}$ in the regret upper bound is replaced by $q_{ti^*}(1 - q_{ti^*})$.
> As $(1 - q_{ti*}) = \sum_{i\neq i^*} q_{ti}$ holds,
> this leads to a regret upper bound expressed in terms of non-optimal arm selection probabilities, making the self-bounding technique applicable.
> Similar techniques are used in [Zimmert+ 2019].
> A description of these intuitions will be added in the revised version.
>
> > Besides suboptimal orders of logarithmic terms, one weakness is that the gap-dependent constants are still not optimal. For example, consider standard bandit feedback. The bound in this case is $\frac{\alpha (\ln T)^3}{\Delta_{\min}}$, which is not ideal once K is very large and there is only one small gap. I think it would be great to comment this somewhere or show some bounds of optimal stochastic algorithms in the table.
>
> In the revised version, the table will also include the regret upper bounds in important special cases including standard bandit settings, and explanations of the gap between the upper and lower bound in special cases will be added.
>
>
> > Can authors talk about whether it is possible to get bounds scaling with $\sum_{i \in S} \frac{1}{\Delta_i}$ for strongly observable graphs under this approach?
>
>
> We think that it is difficult to derive a regret upper bound of form $\tilde{O}(\sum_{i} 1 / \Delta_i)$ as far as the regret analysis is done via the sum of Shannon entropies as in the current approach.
> To overcome this barrier, it may be beneficial to consider a regularization $\sum_{i \in V} \beta_{ti} p_i \ln p_i$ in which the weights $\beta_{ti}$ for each arm are controlled separately instead of the same weight regularization $\beta_{t} \sum_{i \in V} p_i \ln p_i$ for all arms as is currently done.
>
> Reference
>
> - J. Zimmert, H. Luo, and C.-Y. Wei. Beating stochastic and adversarial semi-bandits optimally and simultaneously. In International Conference on Machine Learning, pages 7683–7692. PMLR, 2019.

---

### Meta-Review · Area_Chair_U6U6 · 2022-08-24

**Recommendation:** Accept
**Confidence:** Certain

**Metareview:**

The paper received four reviews from experts in online learning, who all strongly support acceptance.  As summarized very well in the reviews, this is a well-written paper that makes a solid and elegant contribution to the line of work on best-of-both-worlds online learning.  The authors have effectively addressed the (relatively minor) concerns indicated in the reviews.  I wholeheartedly recommend the paper is accepted.

One issue brought in the reviews is the non-trivial overlap with a different paper entitled “A Near-Optimal Best-of-Both-Worlds Algorithm for Online Learning with Feedback Graphs”.  While this situation did not affect the decision in any way, I urge the authors to properly cite this paper in their final version and discuss how it related to their contribution.

**Award:**

No

---

### Decision · Program_Chairs · 2022-09-14

Accept